# Identifying and Understanding Cross-Class Features in Adversarial Training

**Zeming Wei** [1]   **Yiwen Guo** [2]   **Yisen Wang** [3][4]

## Abstract

Adversarial training (AT) has been considered one of the most effective methods for making deep neural networks robust against adversarial attacks, while the training mechanisms and dynamics of AT remain open research problems. In this paper, we present a novel perspective on studying AT through the lens of class-wise feature attribution. Specifically, we identify the impact of a key family of features on AT that are shared by multiple classes, which we call cross-class features. These features are typically useful for robust classification, which we offer theoretical evidence to illustrate through a synthetic data model. Through systematic studies across multiple model architectures and settings, we find that during the initial stage of AT, the model tends to learn more cross-class features until the best robustness checkpoint. As AT further squeezes the training robust loss and causes robust overfitting, the model tends to make decisions based on more class-specific features. Based on these discoveries, we further provide a unified view of two existing properties of AT, including the advantage of soft-label training and robust overfitting. Overall, these insights refine the current understanding of AT mechanisms and provide new perspectives on studying them. Our code is available at https://github.com/PKU-ML/Cross-Class-Features-AT.

## 1. Introduction

As the existence of adversarial examples (Goodfellow et al., 2014) has led to significant safety concerns of deep neural networks (DNNs), a series of methods (Papernot et al.,

2016; Cohen et al., 2019; Chen et al., 2024) for defending against this threat have been proposed. Adversarial training (AT) (Madry et al., 2018), which adaptively adds adversarial perturbations to samples in the training loop, has been considered one of the most effective ways to make the DNNs more robust to adversarial attacks (Athalye et al., 2018).

Given the unique success in improving adversarial robustness and the complex optimization process of AT, several studies have attempted to interpret AT through different perspectives like feature visualization (Ilyas et al., 2019; Bai et al., 2021a; Li et al., 2023) and coverage analysis (Wang et al., 2019). However, there are still a few mysterious properties of AT whose underlying mechanisms remain open research problems. First, AT can lead to a phenomenon known as *robust overfitting* (Rice et al., 2020). During AT, a model may achieve its best test robust error at a certain epoch, but the test robust error will gradually increase in the latter stage of training. By contrast, the training robust error consistently decreases, resulting in a large robust generalization gap. Furthermore, although one-hot labels are usually adequate for standard training, integrating soft-label training methods such as knowledge distillation (Hinton et al., 2015) into AT can significantly improve AT whilst mitigating robust overfitting (Chen et al., 2021) (*e.g.* $41\% \to 48\%$ on CIFAR-10 dataset). However, the reasons why soft labels are typically advantageous for AT remain unclear.

In this paper, we explore the mechanisms of AT and offer a unified understanding of the two properties from a new aspect of class-wise feature attribution. Specifically, we divide the features learned by the model into **cross-class** features and **class-specific** features. The cross-class features are shared among multiple classes in the classification task, *e.g.* the feature *wheels* shared by the *automobile* and *truck* classes in the CIFAR-10 dataset (Krizhevsky et al., 2009). We examine how these features are utilized across various stages of AT. Intriguingly, we observe that at the initial stage, the model gradually learns more cross-class features until reaching the most robust checkpoint. In contrast, at later checkpoints where robust overfitting occurs, the model tends to make decisions based more on class-specific features and decreases its dependence on cross-class features. Furthermore, we find that models trained with properly learned soft labels, like knowledge distillation, can preserve more cross-class features during AT whilst mitigating robust overfitting.

[1]School of Mathematical Sciences, Peking University [2]Independent Researcher [3]State Key Lab of General Artificial Intelligence, School of Intelligence Science and Technology, Peking University [4]Institute for Artificial Intelligence, Peking University. Correspondence to: Yisen Wang <yisen.wang@pku.edu.cn>.

*Proceedings of the $42^{nd}$ International Conference on Machine Learning*, Vancouver, Canada. PMLR 267, 2025. Copyright 2025 by the author(s).

Motivated by these observations, we propose a novel hypothesis of the AT training dynamics. During the initial stage of AT, the model learns both class-specific and cross-class features simultaneously, since these features are both helpful for reducing robust loss (*i.e.*, the cross-entropy loss on adversarial examples) when this loss is large. However, as training progresses and the robust loss decreases to a certain degree, the model begins to abandon cross-class features and makes decisions based mainly on class-specific features, which is caused by cross-class features raising positive logits on other classes and yielding positive robust loss in AT under one-hot labels. Therefore, the model tends to neglect these features to further decrease the robust loss. However, these cross-class features are helpful for robust classification (*e.g.*, a feature shared by classes $y_1, y_2$ helps the model distinguish samples in class $y_1$ from other classes $y_3, \cdots, y_n$), and using only class-specific features is insufficient to achieve the best robust accuracy. We discuss this insight in detail in Section 4. As a result, the robust test accuracy (*i.e.*, the accuracy of the model on adversarial examples) gradually decreases, leading to the robust overfitting issue. In addition, this hypothesis also explains why soft-label training methods typically improve AT as well as alleviate robust overfitting, as their softened labels can preserve more cross-class features during AT than standard one-hot labels.

We provide extensive empirical evidence to support the observations and the hypothesis. First, we propose a metric to measure the usage of the cross-class features for a certain model. Then, among various perturbation norms, datasets, and architectures, we show that the best robustness model consistently uses more cross-class features than the robust overfitted ones, showing a clear correlation between robust generalization and cross-class features. We further provide theoretical insights to intuitively understand this effect through a synthetic data model, where we show that cross-class features are more sensitive to robust loss, but they are indeed helpful for robust classification. Finally, we extend our study to more scenarios, including discussions on larger training perturbation $\epsilon$, alternative metrics, standard training, and fast adversarial training (Wong et al., 2020; Andriushchenko & Flammarion, 2020), to further support our insights.

Our contributions can be summarized as follows:

1. We propose a new hypothesis for the training mechanism in AT from the perspective of cross-class features. Specifically, the model gradually learns them at the initial stage of AT, and tends to reduce the reliance on them after a certain stage. However, these features are actually helpful for robust generalization.

2. We provide both empirical and theoretical evidence to support this understanding. Empirically, we mea-

sure the usage of cross-class features through different stages of AT. We also substantiate these assertions in a synthetic data model with decoupled cross-class and class-specific features.

3. Based on our understanding, we further provide a unified interpretation of some intriguing properties of AT, like robust overfitting and the advantage of soft-label training, substantiating a novel perspective to study AT that warrants further investigation.

## 2. Background and Related Work

### 2.1. Adversarial Training

Adversarial training (AT) (Madry et al., 2018) has been widely recognized as one of the most effective approaches to improving the robustness of models (Athalye et al., 2018), which can be formulated as the following min-max optimization problem:

$$\min_{\boldsymbol{\theta}} \frac{1}{N} \sum_{i=1}^{N} \max_{\|\delta_i\|_p \leq \epsilon} \ell(f(\boldsymbol{\theta}, x_i + \delta_i), y_i), \qquad (1)$$

where $\boldsymbol{\theta}$ represents the model parameter, $\ell$ is the loss function (*i.e.* cross-entropy loss), $(x_i, y_i)$ is the $i$-th sample-label pair in training set, and $\epsilon$ is the perturbation bound. For the inner maximization, Projected Gradient Descent (PGD) (Madry et al., 2018) is generally used to craft the adversarial example:

$$x^{t+1} = \Pi_{\mathcal{B}(x,\epsilon)}(x^t + \alpha \cdot \text{sign}(\nabla_x \ell(\theta; x^t, y))), \quad (2)$$

where $\Pi$ is the function that projects the sample onto an allowed region of perturbation, *i.e.,* $\mathcal{B}(x, \epsilon) = \{x' : \|x' - x\|_p \leq \epsilon\}$, and $\alpha$ controls the step size of gradient ascent. Throughout this thread, numerous variants of AT were proposed from various perspectives, *e.g.,* loss function (Zhang et al., 2019; Wang et al., 2020), computational cost (Shafahi et al., 2019; Wong et al., 2020), and model architecture (Huang et al., 2021; Mo et al., 2022). However, the min-max optimization nature of AT makes its training dynamics a black box, and understanding the internal mechanisms of AT remains an open research area problem (Li & Li, 2024; Wang et al., 2024; Zhang et al., 2024).

### 2.2. Robust Overfitting

Despite success in improving robustness, AT suffers from a problem known as *robust overfitting* (Rice et al., 2020). As shown in Figure 1, the model may perform best on the test dataset at a certain epoch during AT, but in the later stages, the model's performance on the test data gradually worsens. Meanwhile, the model's robust error on the training data continues to decrease, leading to a significant generalization gap in adversarial training. Moreover, for commonly used

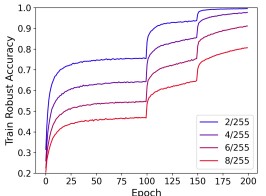 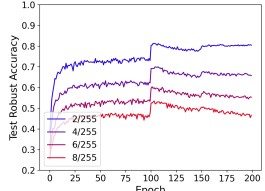

(a) Train robust accuracy    (b) Test robust accuracy

*Figure 1.* Train and test robust accuracy of AT on CIFAR-10 dataset with $\ell_\infty$-norm bound $\epsilon \in \{2, 4, 6, 8\}/255$.

perturbation bound $\epsilon$ (*e.g.* $[0, 8/255]$ for $\ell_\infty$-norm) in AT, a relatively large $\epsilon$ suffers from more severe robust overfitting. By contrast, for a small $\epsilon = 2/255$, this effect is relatively less pronounced. To address the robust overfitting issue in AT, several techniques have been introduced from various perspectives, like data augmentation (Rebuffi et al., 2021; Li & Spratling, 2023) and flatness regularization (Wu et al., 2020; Yu et al., 2022a). Meanwhile, a series of works attempt to interpret the mechanism of robust overfitting through data-wise loss (Yu et al., 2022b) and label noises (Dong et al., 2022a). In this work, we provide a new perspective to refine the current understanding of robust overfitting from class-wise feature analysis.

## 2.3. Adversarial Training with Smoothed Labels

Another intriguing property of AT is the advantage of using properly smoothed labels to replace one-hot labels, *e.g.*, leveraging knowledge distillation (Hinton et al., 2015; Chen et al., 2021) or using temporal ensembling (Laine & Aila, 2017; Dong et al., 2022b; Wang & Wang, 2022). For example, the loss function in Equation (1) of AT with knowledge distillation can be reformulated as

$$\tilde{\ell}(\theta; \theta_0, x + \delta, y) = (1 - \lambda)\ell_{\text{CE}}(f(\theta, x + \delta), y)$$
$$+ \lambda \cdot \text{KL}(f(\theta, x + \delta)/T, f(\theta_0, x + \delta)/T) \quad (3)$$

where $\ell_{\text{CE}}$ is the cross-entropy loss, KL is the Kullback–Leibler divergence, $T$ is the distillation temperature and $\theta_0$ is the teacher model. This type of loss function explicitly converts a one-hot label into a smoothed one, where the model does not necessarily achieve the minimized loss by outputting only a one-hot prediction logit. Motivated by its success in improving AT and mitigating robust overfitting, a series of variants of smoothed-label AT have been proposed (Zhu et al., 2022; Zi et al., 2021; Huang et al., 2023; Yue et al., 2023; Wu et al., 2024), but there is still a lack of a unified view of how they improve the peak performance of AT and also mitigate robust overfitting.

## 3. Cross-Class (Robust) Features

In this section, we elaborate on our proposed understanding of robust overfitting in AT via cross-class features. We first propose a metric of cross-class feature usage for a model in AT. Then, with comprehensive empirical evidence, we demonstrate the dynamics of the model in terms of learning these features during AT, as well as their relationship with robust overfitting and knowledge distillation.

### 3.1. Measuring the Usage of Cross-Class Features

Consider a $K$-class classification task. Let $f(\cdot) = Wg(\cdot)$ represent a classifier, where $g$ is the feature extractor with $n$ dimension and $W \in \mathbb{R}^{K \times n}$ is the linear layer. For a given sample $x$ from the $i$-th class, the output logit for the $i$-th class is

$$f(x)_i = W[i]^T g(x) = \sum_{j=1}^n g(x)_j W[i, j], \quad (4)$$

where $W[i]$ is the $i$-th row of $W$. Intuitively, $g(x)_j W[i, j]$ represents how the $j$-th feature influences the logit of the $i$-th class prediction of $f(x)$. Thus we use

$$A_i(x) = (g(x)_1 W[i, 1], \cdots, g(x)_n W[i, n]) \quad (5)$$

as the *attribution vector* for the sample $x$ on class $i$, where the $j$-th element denotes the weight of the $j$-th feature.

**Characterizing Cross-class Features.** We consider the similarity of attribution vectors. If the attribution vectors of samples $x_1$ and $x_2$ are highly similar, the model tends to use more features shared by them when calculating their logits for their classe (Bai et al., 2021a; Du et al., 2024). On the other hand, if the attribution vectors of $x_1$ and $x_2$ are almost orthogonal, the model uses fewer shared features, or they just do not share features. Further, this observation can be generalized to $K$ classes. We model the feature attribution vector of a given class as the average of the vectors of the test samples in this class. Further, since we only focus on the feature attribution in the context of adversarial robustness, we only consider the usage of **robust features** (Tsipras et al., 2019; Ilyas et al., 2019) for classifying adversarial examples. Thus, we craft adversarial examples and analyze their attributions to measure the usage of shared robust features.

As discussed, we can measure the usage of cross-class robust features shared by different classes with the similarity of their attribution vectors. Therefore, we construct the *feature attribution correlation matrix* using the cosine similarity between the attribution vectors:

$$C[i, j] = \frac{A_i \cdot A_j}{\|A_i\|_2 \cdot \|A_j\|_2}. \quad (6)$$

The complete algorithm of calculating matrix $C$ is shown in Algorithm 1 in Appendix. For two classes indexed by $i$ and $j$, $C[i, j]$ represents the similarity of their feature attribution vector, where a higher value indicates the model uses more features shared by these classes.

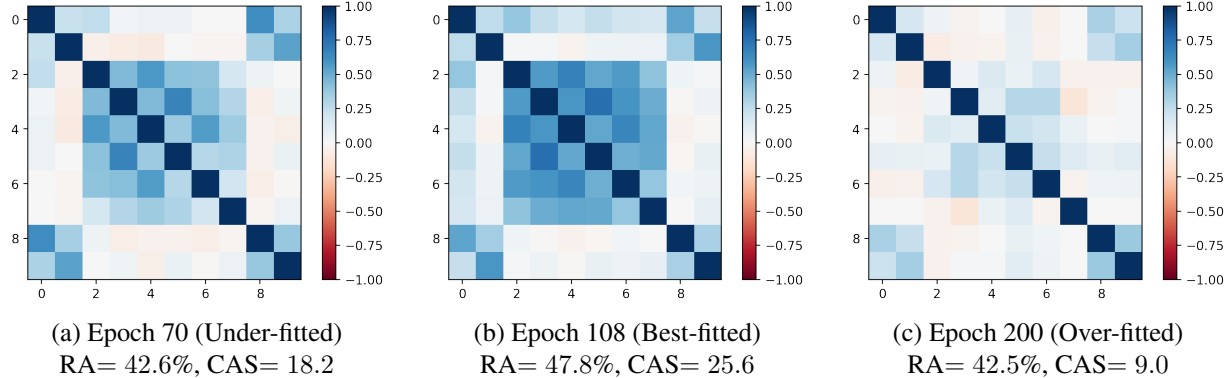

(a) Epoch 70 (Under-fitted)
RA= 42.6%, CAS= 18.2

(b) Epoch 108 (Best-fitted)
RA= 47.8%, CAS= 25.6

(c) Epoch 200 (Over-fitted)
RA= 42.5%, CAS= 9.0

*Figure 2.* Feature Attribution Correlation Matrix of models at different stages in AT, with their test robust accuracy (**RA**) and **CAS**. Class index: airplane (0), automobile (1), bird (2), cat (3), deer (4), dog (5), frog (6), horse (7), ship (8), truck (9).

**Numerical Metric.** To further support our claims, we propose a numerical metric named **C**lass **A**ttribution **S**imilarity (CAS) defined on the correlation matrix $C$:

$$CAS(C) = \sum_{i \neq j} \max(C[i,j], 0) \tag{7}$$

The $\max$ function is used since we only focus on the positive correlations, and the negative elements are small (see Figure 2) and do not affect our analysis. As a numerical indicator, CAS can quantitatively reflect the usage of cross-class features for a certain checkpoint.

### 3.2. Preliminary Study

Based on the proposed measurements, we first visualize the feature attribution correlation matrices of vanilla AT (Madry et al., 2018). For the detailed configurations of training, we follow the implementation of (Pang et al., 2021), which provides a popular repository of AT with basic training tricks. The model is trained on the CIFAR-10 dataset (Krizhevsky et al., 2009) using PreActResNet-18 (He et al., 2016) for 200 epochs, and it achieved its best test robust accuracy at the 108th epoch. A complete list of hyperparameters for experiments in this Section is presented in Appendix B.

**Observations.** As shown in Figure 2, the model demonstrates a fair overlapping effect on feature attribution at the 70th epoch (Under-fitted). Specifically, there are several non-diagonal elements $C[i,j]$ in the correlation matrix $C$ that exhibit a relatively large value (in deeper blue), which indicates that the model leverages more features shared by the classes indexed by $i$ and $j$ when classifying adversarial examples from these two classes. Therefore, the model has already learned several cross-class features in the initial stage of AT. Moreover, when the model achieves its best robustness at the 108th epoch, the overlapping effect on feature attribution becomes clearer, with more non-diagonal elements in $C$ exhibiting larger values. This is also verified by the increase in CAS. However, at the end of AT, where the model is overfitted with decreased test robust accuracy

(RA), the overlapping effect significantly decays, indicating the model substantially neglects cross-class features in its classification. We provide detailed matrices during this training in Figure C in the Appendix.

**Main hypothesis and Robust overfitting.** This intriguing effect motivates us to propose the following hypothesis for the AT mechanism and training dynamics. We identify two kinds of learning mechanisms in AT: (1) Learning *class-specific* features, *i.e.*, the features that are exclusive to only one class; (2) Learning *cross-class features*, *i.e.*, the same or similar features shared by more than one class. For example, the wheels shared by categories *automobile* and *truck*.

Based on this hypothesis, the overall process of AT can be roughly divided into two stages. During the initial phase of AT, the model simultaneously learns exclusive class-wise features and cross-class features. Both of these features help achieve robust generalization and reduce training robust loss. However, once the training robust loss is reduced to a certain degree, it becomes difficult for the model to further decrease it by optimizing cross-class features, since the features shared with other classes tend to raise positive logit on the shared classes. Thus, to further reduce the training robust loss, the model begins to reduce its reliance on cross-class features and places more weight on class-specific features. Meanwhile, due to the strong memorization ability of AT (Dong et al., 2022b), the model also memorizes the training samples along with their corresponding adversarial examples, which further reduces the training robust error. This overall procedure can optimize training robust error but can also hurt test robust error by forgetting cross-class features, leading to a decrease in test robust accuracy and resulting in robust overfitting.

**Soft-label AT.** Our understanding can also explain why soft-label methods, exemplified by knowledge distillation, are helpful for AT in terms of both best checkpoint robustness and mitigating robust overfitting. In the process of AT with knowledge distillation, the teacher model adeptly captures the cross-class features present in the training data, and

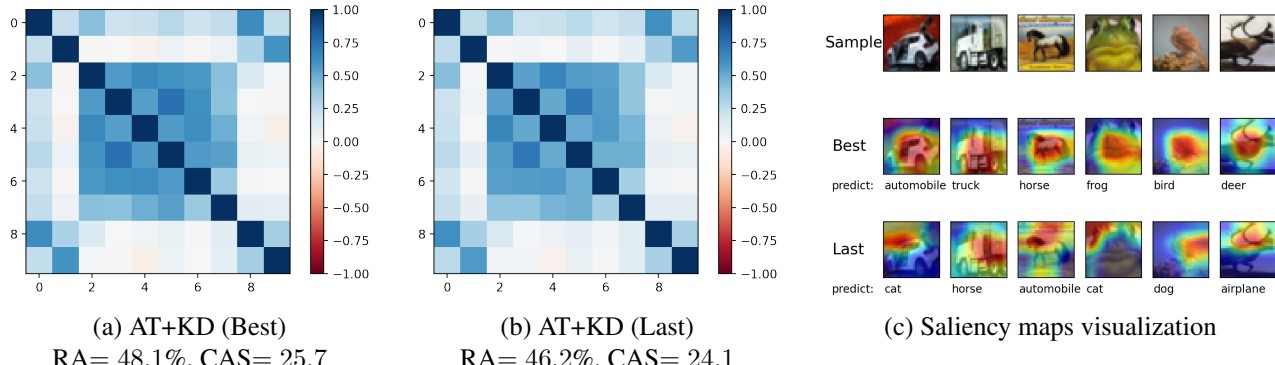

(a) AT+KD (Best)
RA= 48.1%, CAS= 25.7

(b) AT+KD (Last)
RA= 46.2%, CAS= 24.1

(c) Saliency maps visualization

*Figure 3.* (a), (b): matrices for the best and the last checkpoint of AT with knowledge distillation, and their test Robust Accuracy (RA) and CAS. (c): Visualization of saliency map with GradCAM. The top row shows the original sample, and the middle and bottom rows show the saliency map on adversarial examples of the best and the last checkpoint, respectively.

then converts the one-hot label into a more precise one by considering both class-specific and cross-class features. This stands in contrast to vanilla AT with one-hot labels, which primarily emphasize class-specific features and may inadvertently suppress cross-class features in the model weights. Similarly, other smoothed labels, like temporal ensembling, can also effectively mitigate robust overfitting by preserving these crucial features.

To support this claim, we present a comparison between the best and last checkpoint of AT with knowledge distillation in Figure 3 (a) and (b), where no significant differences between the two matrices, nor a large gap between their CAS. Therefore, we conclude that AT with knowledge distillation helps by identifying cross-class features and providing more precise labels by considering these features.

### 3.3. More Empirical Studies

In this section, we conduct more comprehensive studies to support our hypothesis proposed above.

**Visualization of saliency map** To further interpret the concept and role of cross-class features, we present comparisons of the saliency maps on several examples that are correctly classified by the best but misclassified by the last checkpoint under adversarial attack, as shown in Figure 3 (c). The saliency map is derived by Grad-CAM (Selvaraju et al., 2017) on the true labeled classes. Taking the first column as an example, the classes *automobile* and *truck* share similar class-specific discriminative regions (highlighted in the saliency map) like *wheels*. The best checkpoint pays more attention to the overall car including the wheel, whereas the last checkpoint solely focuses on the circular car roof that is exclusive to automobiles. This explains why the last checkpoint misclassifies this sample, for it only identifies this local feature for the true class and does not leverage holistic feature information from the image. The other five samples also exhibit a similar effect, with exclusive fea-

tures being the mane for *horse*, the frog eyes for *frog*, the feather for *bird*, and the antlers for *deer*. Since the final checkpoint makes decisions based only on these limited features, it fails to leverage comprehensive features for classification, making the model more vulnerable to adversarial attacks. More examples on this comparison can be accessed in Appendix D.

**Comparing with different perturbation bound $\epsilon$.** As stated in Section 2, the robust overfitting effect is more severe with larger $\epsilon$ for regular AT $\epsilon$ ($\leq 8/255$), as shown in Figure 1. Intuitively, AT with a larger perturbation bound $\epsilon$ results in a more rigid robust loss. During AT with a large $\epsilon$, cross-class features are more likely to be eliminated by the model to reduce training robust loss, which we prove in Theorem 1 in the next section. In Figure 4, we visualize

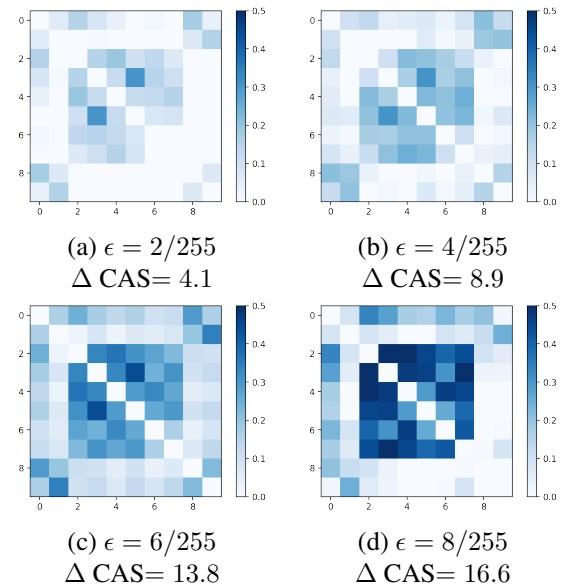

(a) $\epsilon = 2/255$
$\Delta$ CAS= 4.1

(b) $\epsilon = 4/255$
$\Delta$ CAS= 8.9

(c) $\epsilon = 6/255$
$\Delta$ CAS= 13.8

(d) $\epsilon = 8/255$
$\Delta$ CAS= 16.6

*Figure 4.* The **differences** between the feature attribution correlation matrices ($C_{\text{best}} - C_{\text{last}}$) and CAS of the best and the last checkpoint with various training perturbation bound $\epsilon$.

the differences of the feature attribution correlation matrices and CAS between the best and last checkpoint of AT with various perturbation bounds $\epsilon$. The difference between the two matrices indicates how many cross-class features are abandoned by the model from the best checkpoint to the last. When $\epsilon = 2/255$, there is no significant difference between the best and last checkpoint. However, as $\epsilon$ increases, AT exhibits more overfitting effects, and the difference becomes more significant. This also verifies that the forgetting of cross-class features is a key factor of robust overfitting.

Notably, while we mainly focus on AT with practically used $\epsilon$ (e.g., $[0, 8/255]$ for $\ell_\infty$-AT), it is also observed that for extremely large $\epsilon (> 8/255)$, the effect of robust overfitting begins to decline (Wang et al., 2024; Wei et al., 2023). Our interpretation is also compatible with this phenomenon, which we discuss in Section 5.1. In brief, cross-class features are more sensitive under extremely large $\epsilon$, making them even harder to learn at the initial stage of AT. Therefore, even at the best checkpoint, they learn fewer cross-class features, resulting in fewer forgetting of these features in the latter stage of AT.

**More datasets**. We extend our observations by illustrating the comparisons on the **CIFAR-100** (Krizhevsky et al., 2009) and the **TinyImagenet** (mnmoustafa, 2017) datasets in Figure 5. We can see that there are still significant differences between matrices and CAS derived from the best and the last checkpoint of AT on other datasets, showing this effect still holds for various datasets.

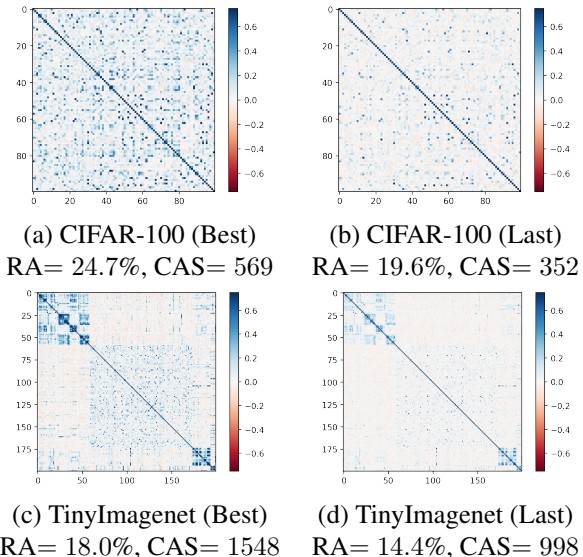

(a) CIFAR-100 (Best)
RA= 24.7%, CAS= 569

(b) CIFAR-100 (Last)
RA= 19.6%, CAS= 352

(c) TinyImagenet (Best)
RA= 18.0%, CAS= 1548

(d) TinyImagenet (Last)
RA= 14.4%, CAS= 998

*Figure 5.* Feature attribution correlation matrices on CIFAR-100 and Tiny-ImageNet datasets. Color bar scaled to $[-0.75, 0.75]$.

$\ell_2$**-norm AT.** We show the comparison of the feature attribution correlation matrices of the best and last checkpoints of $\ell_2$-norm AT ($\epsilon = 128/255$) on CIFAR-10 in Figure 6

(a)(b), where there are still significant differences between matrices from the two checkpoints of $\ell_2$-norm AT. Other training configurations are the same as $\ell_\infty$-norm AT above.

**Transformer architecture.** We show the comparison of the feature attribution correlation matrices of the best and last checkpoints of AT on CIFAR-10 with vision transformer architecture (**Deit-Ti** (Touvron et al., 2021)) in Figure 6 (c)(d). The observation is consistent with other settings.

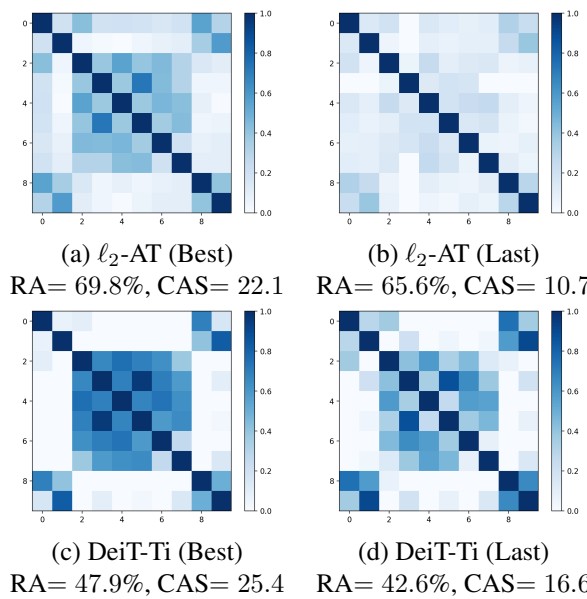

(a) $\ell_2$-AT (Best)
RA= 69.8%, CAS= 22.1

(b) $\ell_2$-AT (Last)
RA= 65.6%, CAS= 10.7

(c) DeiT-Ti (Best)
RA= 47.9%, CAS= 25.4

(d) DeiT-Ti (Last)
RA= 42.6%, CAS= 16.6

*Figure 6.* Feature attribution correlation matrices on $\ell_2$-norm AT and Visual Transformer architecture. Color bar scaled to $[0, 1]$.

Overall, these empirical findings provide a solid justification for our main hypothesis for the learning dynamics of cross-class features during AT. In the following section, we also offer theoretical insights to intuitively understand the role of cross-class features in robust classification.

## 4. Theoretical Insights

In this theoretical framework, we first introduce a synthetic data model and then provide insights into our claims.

### 4.1. Data Distribution and Hypothesis Space

**Data distribution** We consider a tertiary classification task, where each class owns an exclusive feature attribution $x_{E,i}$, and every two classes have a shared cross-class feature attribution $x_{C,j}$. The attribution for each sample can be formulated as $\{x_{E,j}, x_{C,j} | 1 \leq j \leq 3\} \in \mathbb{R}^6$. The data distribution is similar to the model applied in robust and non-robust features (Tsipras et al., 2019), but we only focus on the inner relation between robust features (class-specific or cross-class) and omit the non-robust features.

As discussed above, we model the data distribution of the $i$-th class $y_i$ as $\mathcal{D}_i =:$

$$x_{E,j} \sim \begin{cases} \mathcal{N}(\mu, \sigma^2), & j = i \\ 0, & j \neq i \end{cases}, x_{C,j} \sim \begin{cases} \mathcal{N}(\mu, \sigma^2), & j \neq i \\ 0, & j = i \end{cases} \tag{8}$$

where $i \in \{1, 2, 3\}$, and $\mu, \sigma > 0$. We also assume $\sigma < \sqrt{\pi}\mu$ to control the variance.

**Hypothesis space** We introduce a linear model $f(x)$ in this classification task, which gives $i$-th logit for sample $x$ by $f(x)_i = \sum_j w_{i,j}^E x_{E,j} + \sum_j w_{i,j}^C x_{C,j}$. However, there are 6 parameters in the data samples, making this linear model hard to analyze. Thus we simplify the model based on the following observations. First, we can simply keep $w_{i,j}^E = 0$ for $i \neq j$ and $w_{i,i}^C = 0$ due to the corresponding data distribution is identity to 0. Further, we set $w_{1,1}^E = w_{2,2}^E = w_{3,3}^E = w_1$ and $w_{i,j}^C = w_2 (i \neq j)$ due to symmetry, similar to (Tsipras et al., 2019). Finally, we assume $w_1, w_2 \geq 0$ since $\mu > 0$. Overall, the hypothesis space is $\{f_{\boldsymbol{w}} : \boldsymbol{w} = (w_1, w_2), w_1, w_2 \geq 0\}$ and $f_{\boldsymbol{w}}(x)$ calculates its $i$-th logit by $f_{\boldsymbol{w}}(\boldsymbol{x})_i = w_1 x_{E,i} + w_2 (x_{C,j_1} + x_{C,j_2})$, where $\{j_1, j_2\} = \{1, 2, 3\} \backslash \{i\}$. Now we consider adversarially training $f_{\boldsymbol{w}}$ with $\ell_\infty$-norm perturbation bound $\epsilon < \frac{\mu}{2}$. We also add a regularization term $\frac{\lambda}{2}\|\boldsymbol{w}\|_2^2$ to the overall loss function, which can be modeled as

$$\mathbb{E}_{i \sim \{1,2,3\}} \{\mathbb{E}_{x \sim \mathcal{D}_i} \max_{\|\delta\|_p \leq \epsilon} \ell(w; x + \delta)\} + \frac{\lambda}{2}\|\boldsymbol{w}\|_2^2, \tag{9}$$

where

$$\ell(w; x + \delta) = \max_{\|\delta\|_\infty \leq \epsilon} (\max_{j \neq i} f_{\boldsymbol{w}}(x + \delta)_j - f_{\boldsymbol{w}}(x + \delta)_i). \tag{10}$$

### 4.2. Main results

**Cross-class features are more sensitive to robust loss.** We show that under the robust training loss (10), the model tends to abandon $x_C$ by setting $w_2 = 0$ if $\epsilon$ is larger than a certain threshold. However, any $\epsilon \in (0, \frac{\mu}{2})$ returns a positive $w_1$, as stated in Theorem 1. This result indicates that cross-class features are more sensitive to robust loss and are more likely to be eliminated in AT compared to class-specific features, even when they share the same mean value $\mu$.

**Theorem 1.** *There exists a $\epsilon_0 \in (0, \frac{1}{2}\mu)$, for AT by optimizing the robust loss (10) with $\epsilon \in (0, \epsilon_0)$, the output function obtains $w_2 > 0$; for AT with $\epsilon \in (\epsilon_0, \frac{1}{2}\mu)$, the output function returns $w_2 = 0$. By contrast, AT with $\epsilon \in (0, \frac{1}{2}\mu)$ always obtains $w_1 > 0$.*

This claim is also consistent with our discussion on AT with different $\epsilon$ in Section 3.3. Recall that AT with larger $\epsilon$ tends to compress more cross-class features as shown in Figure 4. This observation can be verified by Theorem 1 that cross-class features are more likely to be eliminated during AT with larger $\epsilon$, which causes more severe robust overfitting.

**Cross-class features are helpful for robust classification.** Although decreasing the value of $w_2$ may reduce the robust training error, we demonstrate in Theorem 2 that using a positive $w_2$ is always more beneficial for robust classification than simply setting $w_2$ to 0.

**Theorem 2.** *For any class $y$, consider weights $w_1 > 0$, $w_2 \in [0, w_1]$, and $\epsilon \in (0, \frac{\mu}{2})$. When sampling $x$ from the distribution of class $y$, increasing the value of $w_2$ enhances the possibility of the model assigning a higher logit to class $y$ than to any other class $y' \neq y$ under adversarial attack. In other words, the probability*

$$\Pr_{x \sim \mathcal{D}_y} [f_w(x + \delta))_y > f_w(x + \delta)_{y'}, \forall \delta : \|\delta\|_\infty \leq \epsilon] \tag{11}$$

*monotonically increases with $w_2$ within the range $[0, w_1]$.*

**Smoothed label preserves cross-class features.** Finally, we show that smoothed labels can help preserve the cross-class features, which justifies why this method can alleviate robust overfitting. Note that due to the symmetry of distributions and weights among classes, we apply label smoothing to simulate knowledge distillation and rewrite the robust loss as

$$\mathbb{E}_{i \sim p_y} \{\mathbb{E}_{x \sim \mathcal{D}_i} \max_{\|\delta\|_p \leq \epsilon} \ell_{\mathrm{LS}}(w; x + \delta)\} + \frac{\lambda}{2}\|\boldsymbol{w}\|_2^2, \tag{12}$$

where $\ell_{\mathrm{LS}}(w; x + \delta)$ is

$$(1 - \beta)[\max_{\|\delta\|_\infty \leq \epsilon} (\max_{j \neq i} f_{\boldsymbol{w}}(x + \delta)_j - f_{\boldsymbol{w}}(x + \delta)_i)]$$
$$- \frac{\beta}{2} \sum_{j \neq i} f_{\boldsymbol{w}}(x + \delta)_j, \tag{13}$$

and $\beta < \frac{1}{3}$ is the interpolation ratio of label smoothing. In Theorem 3 and Corollary 1, we show that not only does the label smoothed loss (13) enable a larger perturbation bound $\epsilon$ for utilizing cross-class features, but also returns a larger $w_2$. This explains that preserving the cross-class features is the reason why smoothed labels help mitigate robust overfitting.

**Theorem 3.** *Consider AT with label smoothing loss (13). There exists an $\epsilon_1 \in (0, \frac{\mu}{2})$ with $\epsilon_1 > \epsilon_0$ derived in Theorem 1, such that for $\epsilon \in (0, \epsilon_1)$, the output function obtains $w_2 > 0$; for $\epsilon \in (\epsilon_1, \frac{1}{2}\mu)$, the output function returns $w_2 = 0$.*

**Corollary 1.** *Let $w_2^*(\epsilon)$ be the value of $w_2$ returned by AT with (10), and $w_2^{LS}(\epsilon)$ be the value of $w_2$ returned by label smoothed loss (13). Then, for $\epsilon \in (0, \epsilon_1)$, we have $w_2^{LS}(\epsilon) > w_2^*(\epsilon)$.*

All proofs can be found in Appendix E. To summarize, our theoretical analysis demonstrates that cross-class features are more sensitive to robust loss, yet helpful for robust classification. We also present a discussion on extension to higher dimensions in Appendix E.5.

# 5. Extended Studies and Discussions

In this section, we extend our observations to broader scenarios to substantiate our understanding.

## 5.1. Regarding extremely large $\epsilon$

Our interpretation is consistent with empirical observations that for commonly used $\epsilon \in [0, 8/255]$, larger perturbation bounds exacerbate robust overfitting. However, it also resolves the seemingly contradictory phenomenon where extremely large $\epsilon$ (e.g., $\epsilon > 8/255$) mitigates overfitting (Wang et al., 2024; Wei et al., 2023). To interpret this phenomenon, recall that our main interpretation for robust overfitting is that the model begins to forget cross-class features after a certain stage. Regarding AT with extremely large $\epsilon$, as we proved in Theorem 1, the more rigid robust loss makes the model even harder to learn cross-class features at the initial stage of AT. Given that fewer cross-class features are learned, the forgetting effect of these features is weakened, thus mitigating robust overfitting. We support this mechanism with empirical validations.

*Table 1.* Comparison of RA and CAS on AT with large $\epsilon$.

| Epoch | 10 | Best | Last |
|---|---|---|---|
| $\epsilon$ for AT | CAS / RA | CAS / RA | CAS / RA |
| 8/255 | 16.7/36.9% | 25.6/47.8% | 9.0/42.5% |
| 12/255 | 15.6/29.8% | 18.9/38.7% | 8.7/34.1% |
| 16/255 | 14.4/23.8% | 17.5/31.3% | 8.4/28.1% |

Specifically, we compare models trained with $\ell_\infty$-norm $\epsilon \in \{8/255, 12/255, 16/255\}$ on CIFAR-10, tracking CAS and robust accuracy across epochs (Table 1). At the 10th epoch, models with $\epsilon = 12/255$ and $16/255$ exhibit lower CAS than $\epsilon = 8/255$, confirming their struggle to learn cross-class features early on. By the best checkpoint, peak CAS values for larger $\epsilon$ remain markedly lower, indicating limited cross-class feature retention. Crucially, the gap in CAS between the best and final checkpoints shrinks as $\epsilon$ increases, mirroring the reduced divergence in robust accuracy. This trend aligns with our hypothesis: extreme $\epsilon$ values suppress cross-class feature acquisition from the outset, leaving fewer features to discard during later stages. Consequently, the attenuated forgetting effect aligns with diminished robust overfitting.

## 5.2. Regarding catastrophic overfitting

Another intriguing property of AT is the catastrophic overfitting phenomenon in fast adversarial training (FAT) (Wong et al., 2020; Andriushchenko & Flammarion, 2020), which applies a single-step perturbation during AT for better efficiency. However, FAT suffers from the catastrophic overfit-

ting issue that the test robust accuracy suddenly decreases to near 0% after a certain epoch (Kim et al., 2021), different from robust overfitting, where the robust accuracy gradually decreases. Our understanding is also compatible with this phenomenon, as discussed in the following.

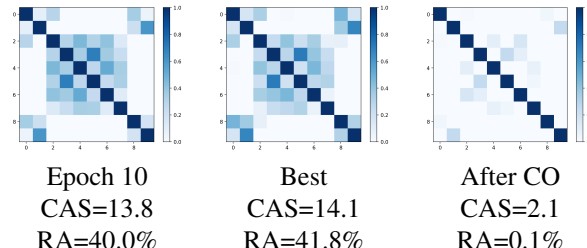

| Epoch 10 | Best | After CO |
|---|---|---|
| CAS=13.8 | CAS=14.1 | CAS=2.1 |
| RA=40.0% | RA=41.8% | RA=0.1% |

*Figure 7.* Feature attribution correlation matrices for **fast adversarial training** at different stages, including epoch 10, best checkpoint, and after catastrophic overfitting (**CO**).

We conduct experiments using the FAT method on the CIFAR-10 dataset, with other settings the same as standard AT. The feature attribution correlation matrices and CAS values at epoch 10, the best checkpoint, and after catastrophic overfitting are presented in Figure 7. Similar to the standard AT, the model has already learned a certain amount of cross-class features at epoch 10, and achieves better robustness and CAS at the best checkpoint. However, after catastrophic overfitting occurs, the CAS value plummets to 2.1, and the robust accuracy drops to near zero. This suggests that during catastrophic overfitting, the model almost completely forgets the cross-class features it had learned earlier. Therefore, the forgetting of cross-class features is also an underlying mechanism of catastrophic overfitting, which aligns well with our observations on robust overfitting.

## 5.3. Instance-wise Metric Analysis

In this section, we provide an alternative metric to further support our claims by calculating the feature attribution matrix and CAS **instance-wisely**. Specifically, when considering classes $i$ and $j$, for each sample $x$ from class $i$, we identify its most similar counterpart $x'$ from class $j$. We then calculate their cosine similarity and average the results over all samples in class $i$. In this context, $x'$ can be interpreted as the sample in class $j$ that shares the most cross-class features with $x$ among all samples in class $j$, which provides another way to quantify the utilization of cross-class features. We also attempt to average over all sample pairs $(x, x')$ in classes $i$ and $j$, but due to high variance among samples, each element in the correlation matrix $C$ hovered near zero throughout all epochs in adversarial training, rendering it unable to provide meaningful information. Based on this metric, we conduct a similar study by calculating the matrices and I-CAS for the best and last checkpoints of $\ell_\infty$ and $\ell_2$-AT, and the results are shown in Figure 8.

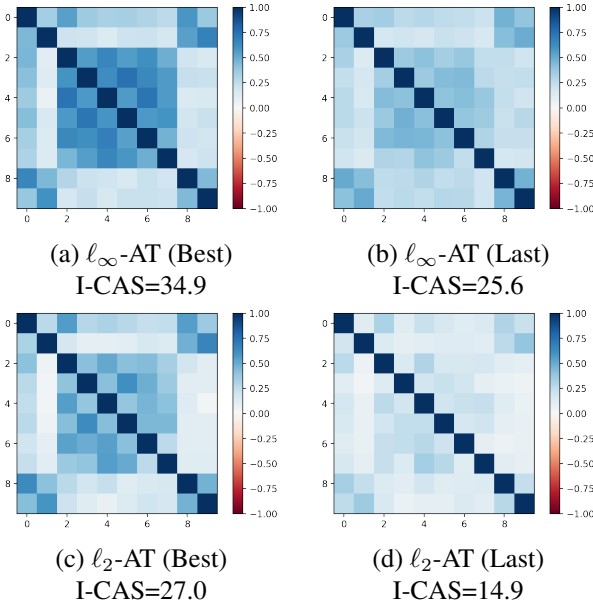

(a) $\ell_\infty$-AT (Best)
I-CAS=34.9

(b) $\ell_\infty$-AT (Last)
I-CAS=25.6

(c) $\ell_2$-AT (Best)
I-CAS=27.0

(d) $\ell_2$-AT (Last)
I-CAS=14.9

*Figure 8.* **Instance-wise** feature attribution correlation matrices.

Consistent with the results for class-wise attribution vectors, it is still observed that there is a significant decrease in the usage of cross-class features from the best checkpoint to the last for both $\ell_\infty$ and $\ell_2$-AT. This observation further substantiates our understanding of cross-class features.

### 5.4. Regarding Standard Training

We also extend our experimental scope to include standard training. The experimental settings are the same as those outlined in previous sections for CIFAR-10, with the only difference being the absence of perturbations in standard training. We present the matrices and CAS results for epochs $\{50, 100, 150, 200\}$ in Figure 9. Considering that standard training only focuses on clean accuracy and exhibits negligible robustness, we calculate the feature attribution vectors using clean examples. These results reveal a clear lack of differences between them, particularly in the latter stages (150th and 200th), where the training tends to converge. This observation is consistent with the characteristic of standard training, which generally does not exhibit significant overfitting (Jiang et al., 2020; Guo et al., 2023). In addition, the numerical magnitude of CAS by these models is significantly lower than AT (generally $> 20$), showing that they just use fewer cross-class features in standard classification.

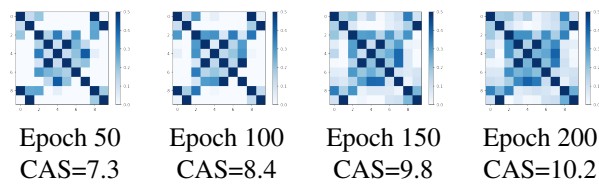

Epoch 50
CAS=7.3

Epoch 100
CAS=8.4

Epoch 150
CAS=9.8

Epoch 200
CAS=10.2

*Figure 9.* Feature attribution correlation matrices for **standard training** at different stages. Color bar scaled to $[0, 0.5]$.

### 5.5. Discussion on future applications

Finally, building on our comprehensive study on the critical role of cross-class features in AT, we discuss their potential future applications in robust generalization research. First, similar to the robust/non-robust feature decomposition (Tsipras et al., 2019), our cross-class feature model has the potential for more in-depth modeling of adversarial robustness, contributing new tools in its theoretical analysis. Meanwhile, for AT algorithmic design, we list some future perspectives of cross-class feature as follows:

- **Data (re)sampling**. While generated data is prone to help advance adversarial robustness (Gowal et al., 2021; Wang et al., 2023), it requires significantly more data and computational costs. From the cross-class feature perspective, adaptively sampling generated data with considerations of cross-class relationships may improve the efficiency of large-scale AT.

- **AT configurations**. Customizing AT configurations like perturbation margins or neighborhoods is useful for improving robustness (Wei et al., 2023; Cheng et al., 2022). In this regard, customizing sample-wise or class-wise AT configurations based on cross-class relationships may further improve robustness.

- **Module design**. The model architecture (Huang et al., 2021) and activation mechanisms (Bai et al., 2021b) are crucial in improving robustness. Thus, designing modules that implicitly or explicitly emphasize cross-class features may enhance robustness.

## 6. Conclusion

In this work, we present a novel perspective to understand adversarial training (AT) dynamics through the lens of cross-class features. We demonstrate that cross-class features, which are shared across multiple classes, play a critical role in achieving robust generalization. However, as training progresses, models increasingly rely on class-specific features to minimize robust training loss, leading to the forgetting of cross-class features and subsequent robust overfitting. Our empirical analyses across datasets, architectures, and perturbation norms, as well as theoretical insights, validate this hypothesis that models at peak robustness utilize significantly more cross-class features than overfitted ones. Furthermore, we reveal that soft-label methods like knowledge distillation mitigate overfitting by preserving cross-class features, aligning with their empirical success. These findings are further substantiated through extended studies like large perturbation AT, fast adversarial training, alternative metrics, and comparison with standard training. Overall, our understanding provides a unified explanation for robust overfitting and the efficacy of label smoothing in AT, offering new insights for studying robust generalization.

## Acknowledgments

Yisen Wang was supported by National Key R&D Program of China (2022ZD0160300), National Natural Science Foundation of China (92370129, 62376010), Beijing Nova Program (20230484344, 20240484642), and BaiChuan AI. Zeming Wei was supported by Beijing Natural Science Foundation (QY24035).

## Impact Statement

This paper refines the current understanding of adversarial training (AT) mechanisms, which could improve the robustness of AI systems in safety-critical applications like autonomous driving and cybersecurity. By identifying the role of cross-class features in AT, our findings may inspire more reliable and generalizable defense strategies against adversarial attacks.

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

## A. Algorithm for calculating the feature attribution correlation matrix

We present the complete algorithm for calculating the feature attribution correlation matrix in Algorithm falg. For each class, we first calculate the feature attribution vectors for each test adversarial sample, then calculate the mean of these vectors as the feature attribution vector of this class. Finally, we calculate the cosine similarity of the vectors as the measure of cross-class feature usage for each pair of two classes.

---

**Algorithm 1** Feature Attribution Correlation Matrix

---

**Input:** A DNN classifier $f$ with feature extractor $g$ and linear layer $W$; **Test** dataset $D = \{D_y : y \in \mathcal{Y}\}$; Perturbation margin $\epsilon$;

**Output:** A correlation matrix $C$ measuring the cross-class feature usage

```
/* Record robust feature attribution                                              */
```
**for** $y \in \mathcal{Y}$ **do**

    $A^y \leftarrow (0, \cdots, 0)$ `/* initialization as a` $n$`-dim vector`            `*/`

    **for** $x \in D_y$ **do**

        $\delta \leftarrow \arg\max_{\|\delta\| \leq \epsilon} \ell_{\text{CE}}(f(x + \delta), y)$ `/* untargeted PGD Attack`    `*/`

        $A^y \mathrel{+}= g(x + \delta) \odot W[y]$ `/* point-wise multiplication`       `*/`

    $A^y \leftarrow A^y / |D_y|$ `/* Average`                            `*/`

**for** $1 \leq i, j \leq |\mathcal{Y}|$ **do**

    $C[i, j] \leftarrow \frac{A^i \cdot A^j}{\|A^i\|_2 \cdot \|A^j\|_2}$ `/* Cosine similarity`       `*/`

**return** $C$

---

## B. Detailed training hyperparameters

A complete list of training hyperparameters for AT models is shown in Table 2. For more implementation details, please refer to our code repository https://github.com/PKU-ML/Cross-Class-Features-AT.

*Table 2.* Hyperparameters for AT

| Parameter | Value |
|---|---|
| Train epochs | 200 |
| SGD Momentum | 0.9 |
| batch size | 128 |
| weight decay | 5e−4 |
| Initial learning rate | 0.1 |
| Learning rate decay | 100-th, 150-th epoch |
| Learning rate decay rate | 0.1 |
| training PGD steps | 10 |
| training PGD step size ($\ell_\infty$) | $\epsilon/4$ ($\epsilon$ is the perturbation bound) |
| training PGD step size ($\ell_2$) | $\epsilon/8$ ($\epsilon$ is the perturbation bound) |

## C. More feature attribution correlation matrices at different epochs

We present more feature attribution correlation matrices at different epochs in Figure 10, and the test robust accuracy is aligned with Figure 1(b) (red line, $\epsilon = 8/255$). From the matrices we can see that at the initial stage of AT (10th - 90th Epochs), the model has already learned several cross-class features, and the overlapping effect of class-wise feature attribution achieves the highest at the 110th epoch among the shown matrices. However, for the later stages, where the model starts overfitting, this overlapping effect gradually vanishes, and the model tends to make decisions with fewer cross-class features.

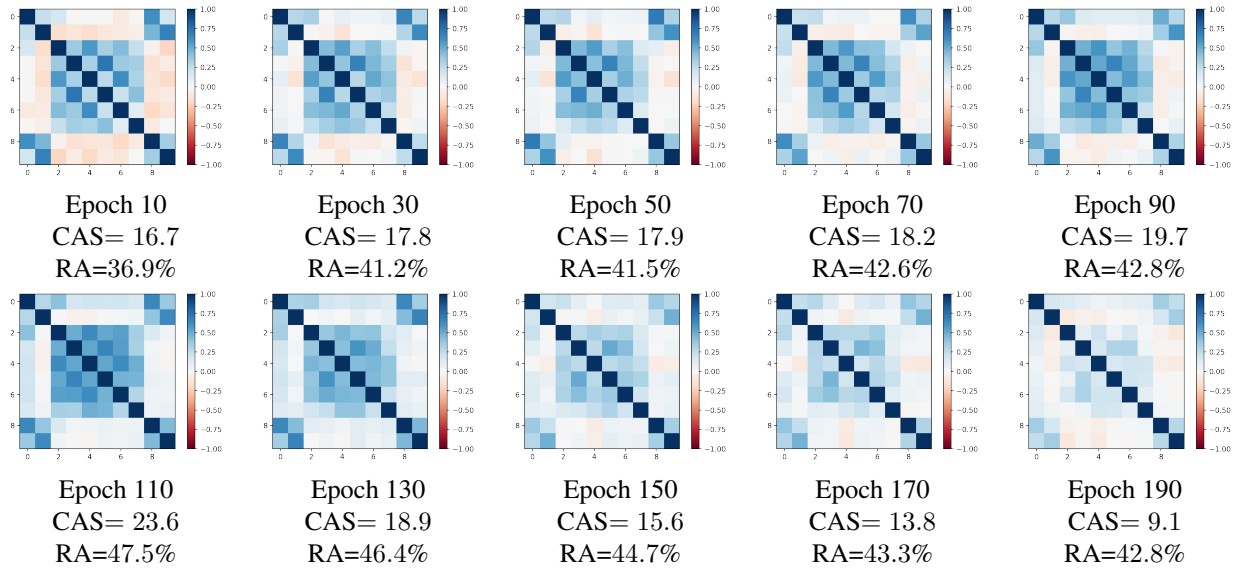

*Figure 10.* Feature attribution correlation matrices, and their corresponding robust accuracy (RA), CAS at different epochs.

## D. More saliency map visualizations

We include more visualization examples (ordered by original sample ID) in Figure 11, where many saliency maps of these examples still exhibit such properties discussed in Section 3.3. However, we acknowledge that not all samples enjoy such clearly interpretable features (*e.g.*, wheels shared by automobiles and trucks), since features learned by neural networks are subtle and do not always align with human intuition, including cross-class features.

## E. Proof of theorems

### E.1. Preliminaries

First, we present some preliminaries and then review the data distribution, the hypothesis space, and the optimization objective.

**Notations**  Let $\mathcal{N}(\mu, \sigma)$ be the normal distribution with mean $\mu$ and variance $\sigma^2$. We denote $\phi(x) = \frac{1}{\sqrt{2\pi}} e^{-\frac{x^2}{2}}$ and $\Phi(x) = \int_{-\infty}^{x} \frac{1}{\sqrt{2\pi}} e^{-\frac{t^2}{2}} \mathrm{d}t = \mathrm{Pr}.(\mathcal{N}(0,1) < x)$ as its probability density function and distribution function.

**Data distribution**  For $i \in \{1, 2, 3\}$, the sample of the $i$-th class is

$$(x_{E,1}, x_{E,2}, x_{E,3}, x_{C,1}, x_{C,2}, x_{C,3}) \in \mathbb{R}^6, \tag{14}$$

follows a distribution

$$\begin{cases} x_{E,j}|(y_i = j) \sim \mathcal{N}(\mu, \sigma^2) \\ x_{E,j}|(y_i \neq j) = 0 \end{cases}, \quad \begin{cases} x_{C,j}|(y_i \neq j) \sim \mathcal{N}(\mu, \sigma^2) \\ x_{C,j}|(y_i = j) = 0 \end{cases}, \tag{15}$$

and $\mu, \sigma > 0$. We also assume $\sigma < \sqrt{\pi}\mu$ to control the variance.

**Hypothesis space**  The hypothesis space is $\{f_{\boldsymbol{w}} : \boldsymbol{w} = (w_1, w_2), w_1, w_2 \geq 0\}$ and $f_{\boldsymbol{w}}(x)$ calculates its $i$-th logit by

$$f_{\boldsymbol{w}}(\boldsymbol{x})_i = w_1 x_{E,i} + w_2(x_{C,j_1} + x_{C,j_2}), \quad \text{where} \quad \{j_1, j_2\} = \{1, 2, 3\} \backslash \{i\}. \tag{16}$$

**Optimization objective**  Consider adversarially training $f_{\boldsymbol{w}}$ with $\ell_\infty$-norm perturbation bound $\epsilon < \frac{\mu}{2}$. We hope that given sample $x \sim \mathcal{D}_i$, under any perturbation $\{\delta : \|\delta\|_\infty \leq \epsilon\}$, the $f(x + \delta)_i$ is larger than any $f(x + \delta)_j$ as much as possible. We also add a regularization term $\frac{\lambda}{2}\|\boldsymbol{w}\|_2^2$ to the loss function.

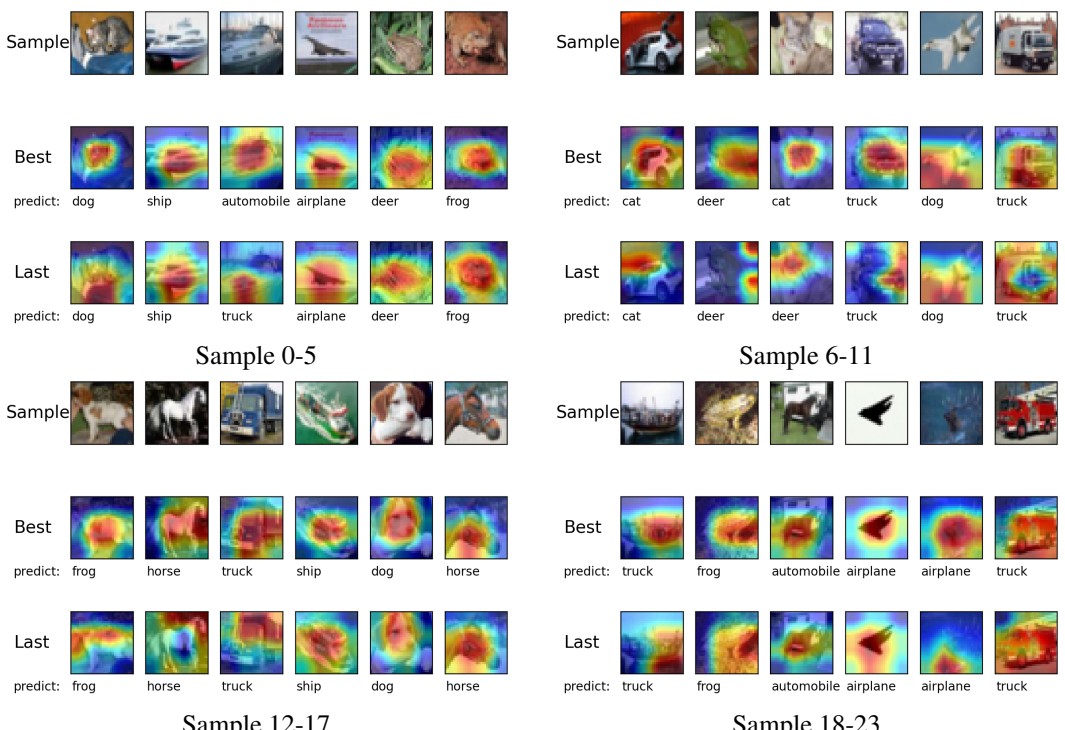

Figure 11. More saliency maps visualization ordered by sample ID in CIFAR-10.

Overall, the loss function can be formulated as

$$\mathcal{L}(f_{\boldsymbol{w}}) = \mathbb{E}_i[\mathbb{E}_{x \sim \mathcal{D}_i} \max_{\|\delta\|_\infty \leq \epsilon} (\max_{j \neq i} f_{\boldsymbol{w}}(x + \delta)_j - f_{\boldsymbol{w}}(x + \delta)_i)] + \frac{\lambda}{2}\|\boldsymbol{w}\|_2^2. \tag{17}$$

### E.2. Proof for Theorem 1

**Theorem 1** *There exists a $\epsilon_0 \in (0, \frac{1}{2}\mu)$, for AT by optimizing the robust loss (17) with $\epsilon \in (0, \epsilon_0)$, the output function obtains $w_2 > 0$; for AT with $\epsilon \in (\epsilon_0, \frac{1}{2}\mu)$, the output function returns $w_2 = 0$. By contrast, AT with $\epsilon \in (0, \frac{1}{2}\mu)$ always obtains $w_1 > 0$.*

To prove Theorem fth:train robust, we need the following lemmas.

**Lemma 1.** *Suppose that $X, Y \sim \mathcal{N}(0, 1)$, and they are independent. Let $Z = \max\{X, Y\}$, then $\mathbb{E}[Z] = \frac{1}{\sqrt{\pi}}$.*

*proof.* Let $p(\cdot)$ and $F(\cdot)$ be the probability density function and distribution function of $Z$, respectively. Then, for any $z \in \mathbb{R}$,

$$F(z) = \Pr(Z < z) = \Pr(\max\{X, Y\} < z) = \Pr(X < z) \cdot \Pr(Y < z) = \Phi^2(z), \tag{18}$$

and we have

$$p(z) = F'(z) = [\Phi^2(z)]' = 2\phi(z)\Phi(z). \tag{19}$$

Thus,

$$
\begin{aligned}
\mathbb{E}[Z] &= \int_{-\infty}^{+\infty} 2z\phi(z)\Phi(z)dz \\
&= 2\int_{-\infty}^{+\infty} z \cdot \frac{1}{\sqrt{2\pi}} e^{-\frac{z^2}{2}} \left(\int_{-\infty}^{z} \frac{1}{\sqrt{2\pi}} e^{-\frac{t^2}{2}} dt\right) dz \\
&= -\frac{1}{\pi} \int_{-\infty}^{+\infty} \left(\int_{-\infty}^{z} e^{-\frac{t^2}{2}} dt\right) d(e^{-\frac{z^2}{2}}) \\
&= -\frac{1}{\pi}[e^{-\frac{z^2}{2}} \int_{-\infty}^{z} e^{-\frac{t^2}{2}} dt]_{-\infty}^{+\infty} + \frac{1}{\pi}\int_{-\infty}^{+\infty} e^{-\frac{z^2}{2}} e^{-\frac{z^2}{2}} dz \\
&= 0 + \frac{1}{\pi}\int_{-\infty}^{+\infty} e^{-z^2} dz = \frac{1}{\sqrt{\pi}}.
\end{aligned}
\tag{20}
$$

**Lemma 2.** *Given* $x = (x_{E,1}, x_{E,2}, x_{E,3}, x_{C,1}, x_{C,2}, x_{C,3}) \sim \mathcal{D}_1$, $\epsilon \in (0, \frac{\mu}{2})$ *and* $\boldsymbol{w} = (w_1, w_2)$, *then* $\delta = (-\epsilon, \epsilon, \epsilon, \epsilon, -\epsilon, -\epsilon)$ *is a solution for* $\delta = \arg \max_{\|\delta\|_\infty \leq \epsilon} [\max_{j \neq 1} f_{\boldsymbol{w}}(x + \delta)_j - f_{\boldsymbol{w}}(x + \delta)_1]$.

*proof.* Denote $\delta = (\delta_{E,1}, \delta_{E,2}, \delta_{E,3}, \delta_{C,1}, \delta_{C,2}, \delta_{C,3})$. Note that for $x \sim \mathcal{D}_1$, we have $x_{E,2} = x_{E,3} = x_{C,1} = 0$. Then,

$$
\begin{aligned}
&\max_{j \neq 1} f_{\boldsymbol{w}}(x + \delta)_j - f_{\boldsymbol{w}}(x + \delta)_1 \\
=& \max_{j \in \{2,3\}} [w_1\delta_{E,2} + w_2\delta_{C,1} + w_2(x_{C,3} + \delta_{C,3}), w_1\delta_{E,3} + w_2\delta_{C,1} + w_2(x_{C,2} + \delta_{C,2})] \\
&- w_1(x_{E,1} + \delta_{E,1}) - w_2(x_{C,2} + \delta_{C,2} + x_{C,3} + \delta_{C,3}) \\
=& w_2\delta_{C,1} + \max_{j \in \{2,3\}} [w_1\delta_{E,2} + w_2(x_{C,3} + \delta_{C,3}), w_1\delta_{E,3} + w_2(x_{C,2} + \delta_{C,2})] \\
&- w_1(x_{E,1} + \delta_{E,1}) - w_2(x_{C,2} + \delta_{C,2} + x_{C,3} + \delta_{C,3}).
\end{aligned}
\tag{21}
$$

Since $w_1, w_2 \geq 0$, it is clear that $\delta_{E,1} = -\epsilon, \delta_{E,2} = \delta_{E,3} = \delta_{C,1} = \epsilon$ are the optimal values for maximizing (21). As for $\delta_{C,2}$ and $\delta_{C,3}$, to prove that $\delta_{C,2} = \delta_{C,2} = -\epsilon$ are the optimal values, by variable simplification ($a' = \delta_{C,2}, b' = \delta_{C,3}$) and dividing by $w_2$ we only need to show that

$$
\max\{a + a', b + b'\} - a' - b' \leq \max\{a - \epsilon, b - \epsilon\} + 2\epsilon
\tag{22}
$$

under the constraint $|a'| \leq \epsilon$ and $|b'| \leq \epsilon$. Note that (22) is equivalent to

$$
\begin{aligned}
&\max\{a + a', b + b'\} - a' - b' \leq \max\{a, b\} + \epsilon \\
\Leftrightarrow& \max\{a + a', b + b'\} \leq \max\{a, b\} + a' + b' + \epsilon \\
\Leftrightarrow& \max\{a + a', b + b'\} \leq \max\{a + a' + b' + \epsilon, b + a' + b' + \epsilon\}.
\end{aligned}
\tag{23}
$$

Since $|b'| \leq \epsilon$, we have $b' + \epsilon \geq 0$ and hence $a + a' \leq a + a' + b' + \epsilon \leq \max\{a + a' + b' + \epsilon, b + a' + b' + \epsilon\}$. Similarly, $b + b' \leq \max\{a + a' + b' + \epsilon, b + a' + b' + \epsilon\}$ and finally we have $\max\{a + a', b + b'\} \leq \max\{a + a' + b' + \epsilon, b + a' + b' + \epsilon\}$. Clearly when $a' = b' = -\epsilon$, the equal sign holds.

**Proof for Theorem 1.** First, due to symmetry, optimizing (17) is equivalent to optimize

$$
\mathbb{E}_{x \sim \mathcal{D}_1}[\max_{\|\delta\|_\infty \leq \epsilon} (\max_{j \neq 1} f_{\boldsymbol{w}}(x + \delta)_j - f_{\boldsymbol{w}}(x + \delta)_1)] + \frac{\lambda}{2}\|\boldsymbol{w}\|_2^2.
\tag{24}
$$

Further, by Lemma flemma:delta we can replace $\delta$ with its optimal value and transform the optimization objective above as

$$
\mathbb{E}_{\hat{x} \sim \hat{\mathcal{D}}_1}(\max_{j \neq i} f_{\boldsymbol{w}}(\hat{x})_j - f_{\boldsymbol{w}}(\hat{x})_i)] + \frac{\lambda}{2}\|\boldsymbol{w}\|_2^2,
\tag{25}
$$

where $\hat{\mathcal{D}}_1$ is the **adversarial data distribution**:

$$\hat{x}_{E,j} \sim \begin{cases} \mathcal{N}(\mu - \epsilon, \sigma^2), & j = 1 \\ \epsilon, & j \neq 1 \end{cases}, \quad \hat{x}_{C,j} \sim \begin{cases} \mathcal{N}(\mu - \epsilon, \sigma^2), & j \neq 1 \\ \epsilon, & j = 1 \end{cases}. \tag{26}$$

Now we calculate the expectation in (25).

$$\mathbb{E}_{\hat{x} \sim \hat{\mathcal{D}}_1}[(\max f_{\boldsymbol{w}}(\hat{x})_j - f_{\boldsymbol{w}}(\hat{x})_i)] + \frac{\lambda}{2}\|\boldsymbol{w}\|_2^2$$

$$= \mathbb{E}_{\hat{x} \sim \hat{\mathcal{D}}_1}[\max(w_1\epsilon + w_2\epsilon + w_2\hat{x}_{C,3}, \ w_1\epsilon + w_2\epsilon + w_2\hat{x}_{C,2}) - w_1\hat{x}_{E,1} - w_2(\hat{x}_{C,2} + \hat{x}_{C,3})] + \frac{\lambda}{2}\|\boldsymbol{w}\|_2^2$$

$$= \mathbb{E}_{\hat{x} \sim \hat{\mathcal{D}}_1}[w_1\epsilon + w_2\epsilon + w_2\max(\hat{x}_{C,3}, \ \hat{x}_{C,2}) - w_1\hat{x}_{E,1} - w_2(\hat{x}_{C,2} + \hat{x}_{C,3})] + \frac{\lambda}{2}\|\boldsymbol{w}\|_2^2 \tag{27}$$

$$= w_1\epsilon + w_2\epsilon + w_2\mathbb{E}_{\hat{x} \sim \hat{\mathcal{D}}_1}[\max(\hat{x}_{C,3}, \ \hat{x}_{C,2})] + \mathbb{E}_{\hat{x} \sim \hat{\mathcal{D}}_1}[-w_1\hat{x}_{E,1} - w_2(\hat{x}_{C,2} + \hat{x}_{C,3})] + \frac{\lambda}{2}\|\boldsymbol{w}\|_2^2$$

$$= w_1\epsilon + w_2\epsilon + w_2\mathbb{E}_{\hat{x} \sim \hat{\mathcal{D}}_1}[\max(\hat{x}_{C,3}, \ \hat{x}_{C,2})] + [-w_1(\mu - \epsilon) - 2w_2(\mu - \epsilon)] + \frac{\lambda}{2}\|\boldsymbol{w}\|_2^2.$$

Finally, since $\hat{x}_{C,3}, \ \hat{x}_{C,2} \sim (\mu - \epsilon, \sigma^2)$ and they are independent, by Lemma flemma:z we have

$$\mathbb{E}[\max(\frac{\hat{x}_{C,3} - (\mu - \epsilon)}{\sigma}, \frac{\hat{x}_{C,2} - (\mu - \epsilon)}{\sigma})] = \frac{1}{\sqrt{\pi}}, \tag{28}$$

hence $\mathbb{E}_{\hat{x} \sim \hat{\mathcal{D}}_1}[\max(\hat{x}_{C,3}, \ \hat{x}_{C,2})] = \mu - \epsilon + \frac{\sigma}{\sqrt{\pi}}$.

Therefore, the optimization objective can be simplified as

$$\mathcal{L}(f_{\boldsymbol{w}}) = (-\mu + 2\epsilon)w_1 + (-\mu + 2\epsilon + \frac{\sigma}{\sqrt{\pi}})w_2 + \frac{\lambda}{2}(w_1^2 + w_2^2). \tag{29}$$

For $w_2$, we have

$$\frac{\partial \mathcal{L}}{\partial w_2} = -\mu + 2\epsilon + \frac{\sigma}{\sqrt{\pi}} + \lambda w_2. \tag{30}$$

Recall that $\sigma < \sqrt{\pi}\mu$. Let $\epsilon_0 = \frac{1}{2}(\mu - \frac{\sigma}{\sqrt{\pi}}) \in (0, \frac{\mu}{2})$. By analyzing the sign of (30), it is clear that for $\epsilon \in (0, \epsilon_0)$, the optimal $w_2$ for minimizing the loss function (29) is

$$w_2 = \frac{\mu - 2\epsilon - \frac{\sigma}{\sqrt{\pi}}}{\lambda}. \tag{31}$$

However, for $\epsilon \in (\epsilon_0, \frac{\mu}{2})$, $\frac{\partial \mathcal{L}}{\partial w_2}$ is always negative, thus the returned $w_2$ by AT is $w_2 = 0$ under the constraint $w_2 \geq 0$.

By contrast,

$$\frac{\partial \mathcal{L}}{\partial w_1} = -\mu + 2\epsilon + \lambda w_1, \tag{32}$$

and for $\epsilon \in (0, \frac{\mu}{2})$, the optimal $w_1$ for minimizing the loss function (29) is always positive:

$$w_1 = \frac{\mu - 2\epsilon}{\lambda} > 0. \tag{33}$$

This ends our proof.

### E.3. Proof for Theorem 2

**Theorem 2** *For any $w_1 > 0$ and $\epsilon \in (0, \frac{\mu}{2})$, if $w_2 \in [0, w_1]$, a larger $w_2$ increases the possibility of the model distinguishing the adversarial examples from any other given class.*

To prove Theorem 2, we need the following lemma.

**Lemma 3.** *Suppose that $X, Y \sim \mathcal{N}(1, \sigma_1^2)$ and they are independent, $\sigma_1 > 0$. Let $Z_t = X + tY$ where $t > 0$. Denote $u(t) = \Pr(Z_t > 0)$, then $u(t)$ is monotonically increasing at $t$ for $t \in [0, 1]$.*

*proof.* Note that $Z_t = X + tY \sim \mathcal{N}(1 + t, (1 + t^2)\sigma_1^2)$. Thus, the distribution function of $Z_t$ is $\Phi_t(z) = \Phi(\frac{z-1-t}{\sqrt{1+t^2}\sigma_1})$, and

$$u(t) = 1 - \Phi_t(0) = 1 - \Phi(\frac{-1-t}{\sqrt{1+t^2}\sigma_1}) = \Phi(\frac{1+t}{\sqrt{1+t^2}\sigma_1}),$$

$$u'(t) = p(\frac{1+t}{\sqrt{1+t^2}\sigma_1})\frac{\sqrt{1+t^2}\sigma_1 - (1+t)\frac{t\sigma_1}{\sqrt{1+t^2}}}{(1+t^2)\sigma_1^2} = p(\frac{1+t}{\sqrt{1+t^2}\sigma_1})\frac{(1+t^2)-(1+t)t}{(1+t^2)\sqrt{1+t^2}\sigma_1} \quad (34)$$

$$= p(\frac{1+t}{\sqrt{1+t^2}\sigma_1})\frac{1-t}{(1+t^2)\sqrt{1+t^2}\sigma_1}.$$

Therefore, for $t \in (0, 1)$, $u'(t) > 0$ and $u(t)$ is monotonically increasing at $t$ for $t \in [0, 1]$.

**Proof for Theorem 2.** Due to symmetry, it's suffice to show that given $w_1$, for $w_2 \in [0, w_1]$, the probability

$$\Pr(f_{\boldsymbol{w}}(\hat{x})_1 > f_{\boldsymbol{w}}(\hat{x})_2), \quad \hat{x} \sim \hat{\mathcal{D}}_1 \quad (35)$$

is monotonically increasing at $w_2$. Note that

$$f_{\boldsymbol{w}}(\hat{x})_1 - f_{\boldsymbol{w}}(\hat{x})_2 = w_1(\hat{x}_{E,1} - \hat{x}_{E,2}) + w_2(\hat{x}_{C,2} - \hat{x}_{C,1}),$$
$$\hat{x}_{E,1} - \hat{x}_{E,2} \sim \mathcal{N}(\mu - 2\epsilon, 2\sigma^2), \quad (36)$$
$$\hat{x}_{C,2} - \hat{x}_{C,1} \sim \mathcal{N}(\mu - 2\epsilon, 2\sigma^2).$$

By dividing $w_1 \cdot (\mu - 2\epsilon)$, and let $t = \frac{w_2}{w_1}$, $X = \frac{\hat{x}_{E,1} - \hat{x}_{E,2}}{\mu - 2\epsilon}$ and $Y = \frac{\hat{x}_{C,2} - \hat{x}_{C,1}}{\mu - 2\epsilon}$, from Lemma flemma:t we know that the probability

$$\Pr(f_{\boldsymbol{w}}(\hat{x})_1 - f_{\boldsymbol{w}}(\hat{x})_2 > 0) \quad (37)$$

is monotonically increasing at $t = \frac{w_2}{w_1}$, and hence increasing at $w_2$. This ends our proof.

### E.4. Proof for Theorem 3 and Corollary 1

**Simplification of knowledge distillation as label smoothing.** In this context, the term 'symmetry' specifically refers to the symmetry of logits for the other two classes when taking the expectation in the loss function (equation 10). When considering data from class $y$, both the distribution of features $x_{E,i}$ and $x_{C_i}$ for the other two classes, as well as their respective weights $w_1$ and $w_2$, exhibit symmetry respectively. Consequently, after applying knowledge distillation, the expectation for logits of the other two classes in the objective loss function becomes identical. To simplify this process, we can employ label smoothing.

We prove Theorem 3 and Corollary 1 in the following. Recall that we define the robust loss under knowledge distillation as

$$\mathcal{L}_{\text{LS}}(f_{\boldsymbol{w}}) = \mathbb{E}_i\{\mathbb{E}_{x \sim \mathcal{D}_i}(1 - \beta)[\max_{\|\delta\|_{\infty} \leq \epsilon} (\max_{j \neq i} f_{\boldsymbol{w}}(x+\delta)_j - f_{\boldsymbol{w}}(x+\delta)_i)] - \frac{\beta}{2}\sum_{j \neq i} f_{\boldsymbol{w}}(x+\delta)_j\} + \frac{\lambda}{2}\|\boldsymbol{w}\|_2^2. \quad (38)$$

**Theorem 3** Consider AT with knowledge distillation loss (38). There exists an $\epsilon_1 > \epsilon_0$, such that for $\epsilon \in (0, \epsilon_1)$, the output function obtains $w_2 > 0$; for $\epsilon \in (\epsilon_1, \frac{1}{2}\mu)$, the output function returns $w_2 = 0$.

**Proof for Theorem 3.** Similar to the proof for Theorem 1, the optimization objective (38) can be simplified as

$$\mathcal{L}_{\text{LS}}(f_{\boldsymbol{w}}) = (1-\beta)[(-\mu+2\epsilon)w_1 + (-\mu+2\epsilon+\frac{\sigma}{\sqrt{\pi}})w_2] - \beta[\epsilon w_1 + \mu w_2] + \frac{\lambda}{2}(w_1^2 + w_2^2)$$
$$= [(1-\beta)\mu + (2-3\beta)\epsilon]w_1 + [(1-\beta)(2\epsilon + \frac{\sigma}{\sqrt{\pi}}) - \mu]w_2 + \frac{\lambda}{2}(w_1^2 + w_2^2). \quad (39)$$

Thus

$$\frac{\mathcal{L}_{\text{LS}}}{w_2} = (1-\beta)(2\epsilon + \frac{\sigma}{\sqrt{\pi}}) - \mu + \lambda w_2, \quad (40)$$

and let $\epsilon_1 = \frac{1}{2}\left(\frac{\mu}{1-\beta} - \frac{\sigma}{\sqrt{\pi}}\right) > \epsilon_0$, similar to the analysis for $\epsilon_0$, we have for $\epsilon \in (0, \epsilon_1)$, the output function obtains $w_2 > 0$; for $\epsilon \in (\epsilon_1, \frac{1}{2}\mu)$, the output function returns $w_2 = 0$. This ends our proof.

**Corollary 1** Let $w_2^*(\epsilon)$ be the value of $w_2$ returned by AT with (17), and $w_2^{\mathrm{LS}}(\epsilon)$ be the value of $w_2$ returned by label smoothed loss (38). Then, for $\epsilon \in (0, \epsilon_1)$, we have $w_2^{\mathrm{LS}}(\epsilon) > w_2^*(\epsilon)$.

**Proof for Corollary 1.** For $\epsilon \in (0, \epsilon_1)$, by analysing the sign of (40), we have

$$w_2^{\mathrm{LS}}(\epsilon) = \frac{\mu - (1-\beta)(2\epsilon + \frac{\sigma}{\sqrt{\pi}})}{\lambda}, \tag{41}$$

and recall that in the proof for Theorem 1 we have

$$w_2^*(\epsilon) = \frac{\mu - (2\epsilon + \frac{\sigma}{\sqrt{\pi}})}{\lambda}, \tag{42}$$

thus it is clear that

$$w_2^{\mathrm{LS}}(\epsilon) - w_2^*(\epsilon) = \frac{\beta(2\epsilon + \frac{\sigma}{\sqrt{\pi}})}{\lambda} > 0. \tag{43}$$

This ends our proof.

### E.5. Extension to higher dimensions

In Section 4, we use a 6-dimensional data representation because it is the smallest dimension for illustrating our insights into cross-class features, since binary classification is incapable of handling cross-class features, as they do not influence the binary classification result. Therefore, we explored a ternary classification problem with minimum dimensions in our framework.

When considering extension to higher dimensions, our theory can be extended to more feature dimensions for ternary classification. For each original feature $x_{E,j}$ or $x_{C,j}$, we can extend them to $x_{E,j}^k$ or $x_{C,j}^k$, where $k = 1, 2, \cdots, K$, thus resulting in $6K$ feature dimensions. Accordingly, we also have corresponding parameters $w_1^k$ and $w_2^k$ for $k = 1, 2, \cdots, K$. Based on this extended model, we can derive similar results in Theorems 1 and 2, where the bounds are set for $w_1^k$ and $w_2^k$. This can be easily derived through calculating the optimal perturbation $\epsilon$ with Lemma 2, calculating the optimizing objectives like Equation (26), and finally deriving the solution of $w_1^k$ and $w_2^k$ for minimizing the objectives like equations (27) and (29).

