# OpenReview forum: "Identifying and Understanding Cross-Class Features in Adversarial Training"
_ICML.cc/2025/Conference — ICML 2025 poster_

### Official Review · Reviewer_RcTN · 2025-03-13

**Overall Recommendation:** 3

**Summary:**

Adversarial Training (AT) is a widely adopted technique for enhancing the robustness of deep learning models against adversarial examples. However, a critical challenge associated with AT is robust overfitting. As training progresses, the robust accuracy on the training set continues to improve, yet the robust accuracy on the test set stops increasing and instead begins to decline.
In this paper, the authors investigate the phenomenon of robust overfitting through the lens of class-wise feature attribution. They observe that the decline in robust accuracy occurs when adversarial training (AT) shifts its focus away from cross-class features and instead relies solely on class-specific features. Based on this observation, they hypothesize that robust overfitting arises due to the reduced reliance on cross-class features during AT. To support their hypothesis, they provide both theoretical analysis and extensive empirical evidence.

**Claims And Evidence:**

I find the authors’ evidence convincing, as they provide solid theoretical proofs for each theorem they propose. Additionally, they validate their findings through extensive experiments conducted across multiple datasets.

**Essential References Not Discussed:**

n/a

**Experimental Designs Or Analyses:**

The authors conduct their experiments on various classification datasets, including CIFAR-10 and CIFAR-100, to visualize feature attribution correlations at different training stages. Their experimental results effectively support their findings, demonstrating that AT tends to reduce reliance on cross-class features as the model becomes overfitted.

**Methods And Evaluation Criteria:**

The metrics and benchmark datasets used in their experiments are appropriate.

**Other Comments Or Suggestions:**

I suggest that the authors provide more detailed explanations of the concepts introduced in their paper. For instance, the terms "robust accuracy" and "robust loss" are used without further clarification. I assume these terms refer specifically to the accuracy and loss measured on adversarial examples, but it would be helpful for the authors to explicitly define them for clarity.

**Other Strengths And Weaknesses:**

The strength of this paper lies in its solid evidence and thorough explanation of why Adversarial Training (AT) suffers from robust overfitting. It also emphasizes the importance of utilizing cross-class features, demonstrating how this approach can help alleviate the issue and improve the effectiveness of AT. However, a concern I have is the paper’s discussion on its significance for future research. While previous work may not have fully understood how AT utilizes class features, we still have an intuitive understanding of the underlying logic. Even with the clearer evidence provided, the challenge remains in how to preserve reliance on cross-class features while continuing to improve the robustness of the model.

**Questions For Authors:**

none

**Relation To Broader Scientific Literature:**

This work primarily explores why traditional AT suffers from robust overfitting and how AT with smooth label can mitigate this issue. By highlighting the importance of preserving cross-class features, the study provides valuable insights that could guide future research toward improving AT and enhancing model robustness.

**Theoretical Claims:**

I have not specifically verified the correctness of their theoretical proofs.

---

> ### Author Rebuttal · Authors · 2025-03-31
>
> Dear Reviewer RcTN,
>
> Thank you for your valuable feedback. We address your concerns below.
>
> ---
>
> **Q1**: A concern I have is the paper’s discussion on its significance for future research. While previous work may not have fully understood how AT utilizes class features, we still have an intuitive understanding of the underlying logic. Even with the clearer evidence provided, the challenge remains in how to preserve reliance on cross-class features while continuing to improve the robustness of the model.
>
> **A1**:  Thank you for the thoughtful comment. We list some potential directions for further applications of our theory as follows:
>
> - **Data (re)sampling**. Since more generated data are prone to be helpful for advancing adversarial robustness [1], it requires significantly more data and computational costs. From the cross-class feature perspective, adaptively sampling generated data **with considerations of class-wise relationship** may improve the efficiency of large-scale AT and decrease the forgetting of cross-class features.
> - **AT configurations**. Customizing AT configurations like perturbation margins or neighborhoods is useful for improving robustness [2]. Since cross-class features are more sensitive against robust loss, customizing AT configurations for different samples or classes **based on class-wise relationships** may mitigate this sensitivity and further improve robustness, as shown in Theorem 1.
> - **Robust module design:** The architecture of a model and the mechanisms of activation play a crucial role in adversarial robustness [3]. Therefore, designing modules that either implicitly or explicitly emphasize cross-class features may enhance robustness. For example, calibrating channel activation can improve robustness [4], thus creating activation mechanisms that preserve more cross-class features can further contribute to this improvement.
>
> In addition to these AT algorithms, we would also like to highlight the **theoretical modeling potential** of our work. Similar to the robust/non-robust feature decomposition [5], which has been applied in many subsequent theoretical works, e.g. [6,7,8], our cross-class feature model has the potential for more in-depth modeling of adversarial robustness, contributing new tools in its theoretical analysis.
>
> [1] Better Diffusion Models Further Improve Adversarial Training. ICML 2023
>
> [2] CAT: Customized Adversarial Training for Improved Robustness. IJCAI 2022
>
> [3] Robust Principles: Architectural Design Principles for Adversarially Robust CNNs. BMVC 2023
>
> [4] Improving Adversarial Robustness via Channel-wise Activation Suppressing. ICLR 2021
>
> [5] Adversarial examples are not bugs, they are features. NeurIPS 2019
>
> [6] On the Tradeoff Between Robustness and Fairness. NeurIPS 2022
>
> [7] Understanding the Impact of Adversarial Robustness on Accuracy Disparity. ICML 2023
>
> [8] Adversarial Training Can Provably Improve Robustness: Theoretical Analysis of Feature Learning Process Under Structured Data. ICLR 2025
>
> ---
>
> **Q2**: I suggest that the authors provide more detailed explanations of the concepts introduced in their paper. For instance, the terms "robust accuracy" and "robust loss" are used without further clarification. I assume these terms refer specifically to the accuracy and loss measured on adversarial examples, but it would be helpful for the authors to explicitly define them for clarity.
>
> **A2**: Thanks for your careful reading. Your assumption is correct, we will clarify these definitions in our revision as follows:
>
> - **Robust accuracy** is the accuracy of the model on adversarial examples.
> - **Robust loss** is the average cross-entropy loss of the model on adversarial examples.
>
> ---
>
> We truly appreciate your valuable and constructive feedback. If you have any further questions or concerns, please let us know.

---

### Official Review · Reviewer_v6k6 · 2025-03-14

**Overall Recommendation:** 3

**Summary:**

This paper explores a unique characteristic of adversarial training from the perspective of class-wise feature attribution. Specifically, it highlights that data often contain **cross-class features**, such as the feature of wheels shared by the automobile and truck classes in the CIFAR-10 dataset.

The authors discover that as training progresses and the robust loss decreases beyond a certain threshold, the model begins to abandon cross-class features. Consequently, the model makes decisions primarily based on class-specific features rather than cross-class ones, which contributes to improved robustness.

Through their experiments, they demonstrate that this phenomenon can be observed across various adversarial training setups.

## update after rebuttal

The insights are valuable and merit inclusion in the main paper, though their earlier presence—particularly the discussion on applications—would have strengthened the submission. Therefore, I maintain my score in favor of acceptance.

**Claims And Evidence:**

The main claim of the paper can be summarized as follows:

**(Main claim)** Cross-class features are ignored after the optimal point during adversarial training, leading to adversarial overfitting.
The main evidence supporting this claim is presented in Figure 2 and Figure 8. Since the paper focuses on the unique characteristics of adversarial training, Figure 8, which illustrates the tendency of standard training, is highly significant. The results demonstrate that adversarial and standard training exhibit different behaviors (so I personally recommend that the authors mention Figure 8 earlier in the paper). Furthermore, the definition of Class Attribution Similarity (CAS) is quite reasonable. The experiments cover various models and datasets, which validates their claim.

I have several questions for the authors:
1) **A more solid explanation for Figure 8 is needed**
In Section 5.3, the authors explain why standard training does not exhibit a similar tendency:  "This observation is consistent with the characteristic of standard training, which generally does not exhibit overfitting." However, as empirically demonstrated in [1], standard training also suffers from overfitting. Please provide a more solid explanation for this observation.

2) **The potential use of cross-class features**
While I acknowledge the importance of this observation, the paper does not provide the benefits of knowledge distillation (or soft-labeling) in terms of adversarial robustness. While they can mitigate the observed phenomenon, it seems they are not effective in improving the robsut accuracy based on Figure 3. Furthermore, class-aware features are difficult to identify, as noted in [2]. Therefore, they cannot be directly used as a regularizer or training trick for adversarial training. How, then, can this insight be leveraged to improve adversarial robustness? If there exists at least a potential direction, it would be valuable to explore these phenomena further.

3) **Regarding catastrophic overfitting**
While the current observation on multi-step adversarial training is quite interesting, catastrophic overfitting is a well-known issue in adversarial training [3-4]. Can the observed phenomenon also be detected in the single-step adversarial training framework?

**Suggestions:**
1) Instead of using numerical labels in all figures, using class labels (e.g., "truck") would provide a more intuitive understanding for readers. As the authors mentioned, automobile (label=1) shows high Class Attribution Similarity (CAS) with truck (label=9). If class names were displayed in the figures, the observations would be clearer.
2) In Equation (5), \( A \) is indexed as \( A_i \), but in Equation (6), it changes to \( A^i \). Please ensure consistency in notation.



- [1] Jiang, Yiding, et al. "Fantastic generalization measures and where to find them." arXiv preprint arXiv:1912.02178 (2019).
- [2] Tsipras, Dimitris, et al. "Robustness may be at odds with accuracy." arXiv preprint arXiv:1805.12152 (2018).
- [3] Wong, Eric, Leslie Rice, and J. Zico Kolter. "Fast is better than free: Revisiting adversarial training." arXiv preprint arXiv:2001.03994 (2020).
- [4] Kim, Hoki, Woojin Lee, and Jaewook Lee. "Understanding catastrophic overfitting in single-step adversarial training." Proceedings of the AAAI Conference on Artificial Intelligence. Vol. 35. No. 9. 2021.

**Essential References Not Discussed:**

N/A

**Experimental Designs Or Analyses:**

Refer to Claims And Evidence.

**Methods And Evaluation Criteria:**

The experiments cover various models and datasets, which validates the observation of the paper.

**Other Comments Or Suggestions:**

N/A

**Other Strengths And Weaknesses:**

N/A

**Questions For Authors:**

Refer to Claims And Evidence.

**Relation To Broader Scientific Literature:**

N/A

**Theoretical Claims:**

I've checked all theoretical claims and verified there is no problem.

---

> ### Author Rebuttal · Authors · 2025-03-31
>
> Dear Reviewer v6k6,
>
> Thank you for your valuable feedback. We address your concerns below.
>
> ---
>
> **Q1**: **A more solid explanation for Figure 8 is needed**
>
> **A1**: Thank you for your thoughtful comment. First, we would like to clarify that Figure 8 aims to show that our theory for adversarial training (AT) is compatible with standard training (ST), where the overfitting issue is far less than robust overfitting in AT. This is supported by the fact that both the accuracy and CAS of ST do not change as significantly as AT in the latter training stage. As for the overfitting in ST stated in [1], it may have other mechanisms like sample memorization [A], but we kindly note that such understandings are not within the scope of our study, since our theory focuses on particular properties of AT. We will add this discussion in our revision.
>
> [A] The Pitfalls of Memorization: When Memorization Hurts Generalization. ICLR 2025
>
> ---
>
> **Q2**: **The potential use of cross-class features**
>
> **A2**: Thanks for the insightful comments. Due to space limitations, please kindly refer to our [response](https://openreview.net/forum?id=FvBYG5jA7k&noteId=ryP80kF51O) to Q1 by Reviewer HgFV for a discussion on the future directions of cross-class features.
>
> ---
>
> **Q3**: **Regarding catastrophic overfitting**
>
> **A3**: Thank you for the insightful comment. Following your suggestion, we implemented a 200-epoch fast adversarial training with 1 step PGD attack, $\ell_\infty$-norm $\epsilon=8/255$ on PreActResNet-18 (same as the main observation experiment in our paper). The CAS for different stages is presented as follows:
>
> | Epoch | 50 | 100 | 150 | 200 |
> | --- | --- | --- | --- | --- |
> | CAS | 16.4 | 18.5 | 7.3 | 2.8 |
> | Robust Accuracy (PGD-10) % | 38.1 | 40.9 | 27.4 | 0.0 |
>
> The results also validate that the forgetting of cross-class features (measured by CAS) can be detected in the catastrophic overfitting of fast adversarial training, which aligns with the main discovery of our paper. We will include these results and related code, figures in our revision. Thanks again for raising this point!
>
> ---
>
> **Q4**: Class names displayed in the figures
>
> **A4**: Thanks for the kind suggestion. We will add these class labels to the saliency maps in our revision.
>
> ---
>
> **Q5**: Please ensure consistency in notation.
>
> **A5**: Thanks for the careful reading. We will unify them as $A_i$ in our revision.
>
> ---
>
> We truly appreciate your valuable and constructive feedback. If you have any further questions or concerns, please let us know.

---

> > ### Comment · Reviewer_v6k6 · 2025-04-09
> >
> > Thank you to the authors for their detailed response.
> > I have reviewed the additional experiments and discussions provided in response to the reviewers' comments. Overall, I appreciate the authors' thoughtful engagement with the feedback. In particular, the extended discussions—such as the Response to Q1 by Reviewer HgFV and the section Regarding Catastrophic Overfitting—significantly strengthen the contribution of the paper. I believe these insights are essential and should be included in the main paper.
> >
> > However, it would have been even more impactful if the discussion on the potential applications of the proposed method (Response to Q1 by Reviewer HgFV) had been included in the initial submission. Doing so could have further advanced the community’s understanding of adversarial robustness.
> >
> > For these reasons, I will maintain my current score.

---

> > > ### Author Response · Authors · 2025-04-09
> > >
> > > Dear Reviewer v6k6,
> > >
> > > Thank you for your further response! We truly appreciate your acknowledgment that our rebuttal significantly strengthens our paper's contribution. We will definitely incorporate all of these extended discussions into the camera-ready version of our paper if accepted, especially the section regarding Catastrophic Overfitting and the future potential of cross-class features.
> > >
> > > Thank you once again for your suggestions, which are invaluable for strengthening the contribution of our work.
> > >
> > > Sincerely,
> > >
> > > Submission 6293 Authors

---

### Official Review · Reviewer_oCW6 · 2025-03-19

**Overall Recommendation:** 4

**Summary:**

While successful at defending models against adversarial examples, the dynamics of adversarial training (AT) are poorly understood. This paper attempts to explain two properties of AT: robust overfitting, and the utility of soft labels over one-hot labels. These properties are studied through the lens of cross-class features, which are features used by the model for classification which are shared by multiple classes. It is shown that a well-fit robust model displays significant correlations between features for different classes; however, when the model is overfit, these cross-class features largely disappear. It is hypothesized that this contributes to the increase in test loss that is characteristic of robust overfitting. Furthermore, it is hypothesized that soft-label methods like knowledge distillation prevent the loss of cross-class features and therefore mitigate robust overfitting. Additional experimental results are provided to support these hypotheses. A theoretical model for adversarial training is thes described which illustrates the utility of cross-class features in robust classification.

**Claims And Evidence:**

I think the major claims made in this paper are backed up by the evidence presented.

**Essential References Not Discussed:**

I am not aware of essential references that are not discussed here.

**Experimental Designs Or Analyses:**

I think the experimental design and analyses presented here are sound. Multiple datasets, architectures, adversarial attacks, and training methods are tested. However, many implementation details are missing, and I don't see a link to a repository with code, so it might be difficult to replicate the experiments.

**Methods And Evaluation Criteria:**

I think the methods and evaluation are largely consistent with the goals of the paper, but some minor issues I have are included in the strengths and weaknesses section below.

**Other Comments Or Suggestions:**

The paper is well written, I didn't find any typos.

**Other Strengths And Weaknesses:**

Strengths
- The description of cross-class features is novel and, according to the experimental results presented, appears to at least partially explain the phenomenon of robust overfitting.
- The feature attribute correlation matrices are a very clear visualization of the phenomenon being described.
- The hypotheses presented are backed up by both experimental and theoretical results.
- The insights from this paper could help inspire further advances in adversarial training. For example, new knowledge distillation methods could be designed that better encourage the maintenance of cross-class features.

Weaknesses
- I find the saliency map visualizations to be unconvincing. It's not clear from figure 3 that these are not cherry-picked examples, and it's difficult to assess the rigor of the claims made in that paragraph.
- The theoretical data model in Section 4 is fixed to 6 dimensions. While the results support the hypotheses posed in previous sections, there isn't any justification as to why we would expect these results to hold for different dimensions.
- The term "feature" feels overloaded in different sections of this paper. In section 3.1, features are defined to be the activations of a feature extractor, while in the saliency map experiments a feature is some abstract quality of the input (i.e. the wheels on a car), while in section 4 a feature is a column in the dataset. I think it would benefit the paper to have a concrete definition of what a feature is in this context (and in particular, whether features are model-dependent).

**Questions For Authors:**

1. What was the reason for choosing a 6-dimensional data representation? At the moment, I'm assuming it's just because that's the smallest dimension with an interesting cross-class feature structure.
2. Would any complications arise when trying to extend the theoretical results to higher dimensions?

**Relation To Broader Scientific Literature:**

This paper studies the problem of adversarial training, which is the focus of a large body of work, and looks specifically at the phenomenon of robust overfitting. This paper connects robust overfitting to the model's use of robust features, discussed in Tsipras et al. and Ilyas et al., and makes use of feature similarity and visualization techniques to explore this connection.

**Theoretical Claims:**

I didn't read the proofs in detail, but I did not find any obvious errors.

---

> ### Author Rebuttal · Authors · 2025-03-31
>
> Dear Reviewer oCW6,
>
> Thank you for your valuable feedback. We address your concerns below.
>
> ---
>
> **Q1**:  Implementation details
>
> **A1**: Thank you for your careful reading. We are committed to publishing our code upon publication. For implementation details regarding model training, we utilize the default hyperparameters according to a well-known adversarial training repository [1] listed below:
>
> | Parameter | |
> | --- | --- |
> | Train epochs | 200 |
> | SGD Momentum | 0.9 |
> | weight decay | $5\times 10^{-4}$ |
> | initial learning rate | 0.1 |
> | learning rate decay | 100, 150-th epochs (decay rate=0.1) |
> | training adversary | 10-step PGD |
>
> We will make these details clear in our revision. Thanks again for the reminder!
>
> [1] Bag of Tricks for Adversarial Training, ICLR 2021, https://github.com/P2333/Bag-of-Tricks-for-AT.
>
> ---
>
> **Q2**: Saliency map visualizations.
>
> **A2**: Thank you for your thoughtful comment. To show these examples are not cherry-picked, we will include a full page of visualization examples (ordered by original sample ID) in the appendix of our revision, where many saliency maps of these examples still exhibit such properties. However, we acknowledge that **not all** samples enjoy such clearly interpretable features (e.g., *wheels* shared by *automobiles* and *trucks*), since features learned by neural networks are subtle and do not always align with human intuition, including cross-class features. Thus, we will tone down related claims made in that paragraph and highlight that the saliency maps are only presented to help understand the concept of cross-class features.
>
> ---
>
> **Q3**: The theoretical data model in Section 4 is fixed to 6 dimensions. What was the reason for choosing a 6-dimensional data representation? Would any complications arise when trying to extend the theoretical results to higher dimensions?
>
> **A3**: Thanks for the thoughtful comment. First, we acknowledge that using a  6-dimensional data representation is the smallest dimension for illustrating our insights into cross-class features, since binary classification is not capable of handling cross-class features as they do not influence the binary classification result. Therefore, we explored a ternary classification problem with minimum dimensions in our framework.
>
> Following your suggestion, we attempted to extend our theory to more feature dimensions for ternary classification . For each original feature $x_{E,j}$ or $x_{C,j}$, we can extend them to $x_{E,j}^k$ or $x_{C,j}^k$, where $k=1,2,\cdots,K$, thus resulting in $6K$ feature dimensions. Accordingly, we also have corresponding parameters $w_1^k$ and $w_2^k$ for $k=1,2,\cdots, K$. Based on this extended model, we can derive similar results in Theorems 1-3, where the bounds are set for $w_1^k$ and $w_2^k$. The proof sketch includes deriving the optimal perturbation $\epsilon$ with Lemma 2, calculating the optimizing objectives like equation (26), and finally derive the solution of $w_1^k$ and $w_2^k$ for minimizing the objectives like equation (27) and (29). Due to space limitations, we are unable to present all proof details here, but we will include this extension in the appendix of our revision. Thank you again for the valuable suggestion!
>
> ---
>
> **Q4**: The term "feature" feels overloaded in different sections of this paper.
>
> **A4**: Thank you for your careful reading. We clarify the definition of “features” as the activation in a particular layer (feature extractor), as stated in Section 3.1. Thus, the feature is model-dependent under this definition.
>
> Regarding saliency maps and Section 4, we apologize for the potential confusion and will replace the term “features” as follows. For saliency maps, we will use the term “class-specific discriminative regions” that was used in the original paper of [2] GradCAM for the highlighted regions. For $x_{E,j}$ and $x_{C,j}$ in Section 4, we will define them as “attribution” instead of features to distinguish them. Thanks again for your kind reminder, we will revise these terms in our revision.
>
> [2] Grad-CAM: Visual Explanations from Deep Networks via Gradient-based Localization. IJCV 2019
>
> ---
>
> We truly appreciate your valuable and constructive feedback. If you have any further questions or concerns, please let us know.

---

### Official Review · Reviewer_HgFV · 2025-03-21

**Overall Recommendation:** 4

**Summary:**

This paper proposed a novel perspective to understand adversarial training. By splitting features into cross-class features and class-specific features and investigating model learning behaviors on cross-class features, this paper demonstrated the importance of cross-class features in improving model robustness. Based on that, this paper also provided an interpretation of robust overfitting and the advantage of soft-label in adversarial training based on cross-class features.

**Claims And Evidence:**

The main hypothesis of this work is that cross-class features are very helpful for improving robust generation of model trained by adversarial training. However, during adversarial training, the model only learns cross-class features at the initial stage and gradually ignores these features after some checkpoint. Authors conduced both theoretically analysis and empirically studies to support this hypothesis.

**Essential References Not Discussed:**

No.

**Experimental Designs Or Analyses:**

Both theoretical analysis and empirical studies strongly support authors' hypothesis about the role of cross-class features played in adversarial training.

**Methods And Evaluation Criteria:**

This work utilized benchmark datasets to conduct empirical studies and proposed a new metric Class Attribution Similarity (CAS) to evaluate the usage of cross-class features during adversarial training.

**Other Comments Or Suggestions:**

No.

**Other Strengths And Weaknesses:**

Pros:
1. This work provided a novel perspective to understand adversarial training and presented a hypothesis about it by investing cross-class features.
2. Both theoretical analysis and empirical studies strongly support authors' hypothesis and also explain the robust overfitting problem and the advantage of soft-label in adversarial training.

Cons:
1. It would be better if authors could discuss, based on the findings in this work, any possible ways to develop advanced adversarial training methods.

**Questions For Authors:**

1. As authors' claimed, after some checkpoint, the trained model reduce its reliance on cross-class features, which result in the reduced robustness on test data; so could we understand this phenomenon in another way, i.e. some class-specific features are naturally contradicted with cross-class features, as cross-class features are helpful for model robustness, the robustness of model will be reduced if model starts learning those contradicted class-specific features after some checkpoint.

**Relation To Broader Scientific Literature:**

This work provided a novel perspective to understand adversarial training, which might be able to inspire future works to develop more advanced adversarial training based method to improve model robustness.

**Theoretical Claims:**

I quickly went through proofs provided in the supplementary material.

---

> ### Author Rebuttal · Authors · 2025-03-31
>
> Dear Reviewer HgFV,
>
> Thank you for your valuable feedback. We address your concerns below.
>
> ---
>
> **Q1**: It would be better if authors could discuss, based on the findings in this work, any possible ways to develop advanced adversarial training methods.
>
> **A1**: Thank you for the thoughtful comment. We list some potential directions for further applications of our theory as follows:
>
> - **Data (re)sampling**. Since more generated data is prone to be helpful for advancing adversarial robustness [1], it requires significantly more data and computational costs. From the cross-class feature perspective, adaptively sampling generated data **with considerations of class-wise relationship** may improve the efficiency of large-scale AT and decrease the forgetting of cross-class features.
> - **AT configurations**. Customizing AT configurations like perturbation margins or neighborhoods is useful for improving robustness [2]. Since cross-class features are more sensitive against robust loss, customizing AT configurations for different samples or classes **based on class-wise relationships** may mitigate this sensitivity and further improve robustness, as shown in Theorem 1.
> - **Robust module design:** The architecture of a model and the mechanisms of activation play a crucial role in adversarial robustness [3]. Therefore, designing modules that either implicitly or explicitly emphasize cross-class features may enhance robustness. For example, calibrating channel activation can improve robustness [4], thus creating activation mechanisms that preserve more cross-class features can further contribute to this improvement.
>
> In addition to these AT algorithms, we would also like to highlight the **theoretical modeling potential** of our work. Similar to the robust/non-robust feature decomposition [5], which has been applied in many subsequent theoretical works, e.g. [6,7,8], our cross-class feature model has the potential for more in-depth modeling of adversarial robustness, contributing new tools in its theoretical analysis.
>
> [1] Better Diffusion Models Further Improve Adversarial Training. ICML 2023
>
> [2] CAT: Customized Adversarial Training for Improved Robustness. IJCAI 2022
>
> [3] Robust Principles: Architectural Design Principles for Adversarially Robust CNNs. BMVC 2023
>
> [4] Improving Adversarial Robustness via Channel-wise Activation Suppressing. ICLR 2021
>
> [5] Adversarial examples are not bugs, they are features. NeurIPS 2019
>
> [6] On the Tradeoff Between Robustness and Fairness. NeurIPS 2022
>
> [7] Understanding the Impact of Adversarial Robustness on Accuracy Disparity. ICML 2023
>
> [8] Adversarial Training Can Provably Improve Robustness: Theoretical Analysis of Feature Learning Process Under Structured Data. ICLR 2025
>
> ---
>
> **Q2**: As authors' claimed, after some checkpoint, the trained model reduce its reliance on cross-class features, which result in the reduced robustness on test data; so could we understand this phenomenon in another way, i.e. some class-specific features are naturally contradicted with cross-class features, as cross-class features are helpful for model robustness, the robustness of model will be reduced if model starts learning those contradicted class-specific features after some checkpoint.
>
> **A2**: Thank you for the alternative interpretation of our theory, which aligns well with our understanding. There may indeed be some contradiction between cross-class and class-specific features, caused by various factors such as model capacity or feature overlap. As a result, learning more class-specific features for lower training robust loss can lead to a decrease in cross-class features, as discussed in Section 3.2. Thank you once again for your insightful comment; we will incorporate this discussion into our revision.
>
> ---
>
> We truly appreciate your valuable and constructive feedback. If you have any further questions or concerns, please let us know.

---

### Decision · Program_Chairs · 2025-05-01

**Decision:**

Accept (poster)

**Comment:**

This paper introduces a novel perspective on adversarial training by identifying and analyzing cross-class features—features shared across multiple classes—and their role in robust model behavior. The study combines theoretical insights with comprehensive empirical evaluations, showing that models rely on cross-class features early in training but tend to abandon them in later stages, contributing to robust overfitting. Reviewers found the hypothesis compelling and well-supported, with strengths including clear visualizations, a new attribution metric (CAS), and relevance to known AT phenomena such as soft-label benefits. Some concerns were raised about practical applicability, overloaded terminology (particularly around "features"), and the interpretability of saliency maps. In the rebuttal, the authors addressed these points effectively by clarifying definitions, outlining concrete application directions, and extending experiments to single-step AT to show generality. Given the conceptual novelty, strong empirical support, and thoughtful engagement with reviewer feedback, the overall recommendation is to accept. To strengthen the final version, the authors are encouraged to improve clarity in terminology, integrate practical applications into the main text, and ensure visual aids are more intuitive.